# Structural bases of IMiD selectivity that emerges by 5-hydroxythalidomide

Hirotake Furihata[1,4], Satoshi Yamanaka [2,4], Toshiaki Honda[3], Yumiko Miyauchi[1], Atsuko Asano[1], Norio Shibata [3], Masaru Tanokura [1✉], Tatsuya Sawasaki [2✉] & Takuya Miyakawa [1✉]

Thalidomide and its derivatives exert not only therapeutic effects as immunomodulatory drugs (IMiDs) but also adverse effects such as teratogenicity, which are due in part to different C2H2 zinc-finger (ZF) transcription factors, IKZF1 (or IKZF3) and SALL4, respectively. Here, we report the structural bases for the SALL4-specific proteasomal degradation induced by 5-hydroxythalidomide, a primary thalidomide metabolite generated by the enzymatic activity of cytochrome P450 isozymes, through the interaction with cereblon (CRBN). The crystal structure of the metabolite-mediated human SALL4-CRBN complex and mutagenesis studies elucidate the complex formation enhanced by the interaction between CRBN and an additional hydroxy group of (S)-5-hydroxythalidomide and the variation in the second residue of β-hairpin structure that underlies the C2H2 ZF-type neo-morphic substrate (neosubstrate) selectivity of 5-hydroxythalidomide. These findings deepen our understanding of the pharmaceutical action of IMiDs and provide structural evidence that the glue-type E3 ligase modulators cause altered neosubstrate specificities through their metabolism.

[1] Department of Applied Biological Chemistry, Graduate School of Agricultural and Life Sciences, The University of Tokyo, 1-1-1 Yayoi, Bunkyo-ku, Tokyo 113-8657, Japan. [2] Proteo-Science Center, Ehime University, 3 Bunkyo-cho, Matsuyama, Ehime 790-8577, Japan. [3] Department of Nanopharmaceutical Sciences and Department of Life Science and Applied Chemistry, Nagoya Institute of Technology, Gokiso-cho, Showa-ku, Aichi 466-8555, Japan. [4]These authors contributed equally: Hirotake Furihata, Satoshi Yamanaka. ✉email: amtanok@mail.ecc.u-tokyo.ac.jp; sawasaki@ehime-u.ac.jp; atmiya@mail.ecc.u-tokyo.ac.jp

Thalidomide, α-(N-phthalimido) glutarimide, was initially used in pregnant women as a sedative drug, which has severe teratogenicity in many tissues and organs, including limb teratogenicity represented as amelia and phocomelia in infants[1–3]. Since it has been demonstrated that thalidomide and its derivatives, lenalidomide and pomalidomide, have immunomodulatory activity and anti-proliferative effects on several hematological cancers, they are widely applied for the therapy of multiple myeloma and other hematologic malignancies as immunomodulatory drugs (IMiDs) despite their adverse effects[4–7]. The direct target of thalidomide and related IMiDs was identified as cereblon (CRBN)[8,9], which was initially reported as the gene associated with inherited autosomal recessive mental retardation[10]. CRBN is the substrate-binding component of the cullin 4 (Cul4)-RING E3 ubiquitin ligase (CRL4) complex[8,9], in which CRBN assembles with the Cul4 scaffold with a RING box protein 1 that acts as an adaptor for the E2 protein through damaged DNA-binding protein 1 (DDB1)[11–14]. The neomorphic E3 ligase activity of CRL4$^{CRBN}$ is induced by the binding of IMiDs to CRBN, and CRBN deficiency in a human multiple myeloma cell line confers resistance to IMiDs[8,9,15]. Thus, the anti-proliferative and immunomodulatory effects of IMiDs are mediated by the CRL4$^{CRBN}$-dependent ubiquitination of neomorphic substrates (neosubstrates) followed by their proteasomal degradation.

The molecular mechanism underlying the therapeutic action of IMiDs has been further elucidated by the identification of the transcription factors, Ikaros (IKZF1) and Aiolos (IKZF3), and casein kinase 1α (CK1α) as neosubstrates for the CRL4$^{CRBN}$ complex[16–19]. IKZF1 and IKZF3 regulate hematological differentiation with multiple Cys$_2$-His$_2$ (C2H2) zinc-finger (ZF) domains[20,21], and CK1α is related to 5q deletion-associated hematopoietic stem cells of myelodysplastic syndrome[19,22]. CRBN recruits IKZF1 or IKZF3 to CRL4$^{CRBN}$ through its interaction with the second C2H2 ZF domain (ZF2), which is mediated by thalidomide and its derivatives[11,23,24]. There are ~800 C2H2 ZF-containing proteins predicted in the human genome[25]. Among the products encoded, 29 ZF domains, including IKZF1 ZF2 and IKZF3 ZF2, have been identified as degrons with the capacity to bind pomalidomide-engaged CRBN[24]. Recently, spalt-like transcription factor 4 (SALL4), PLZF, and p63 have been implicated as teratogenic candidates of IMiDs[26–29]. Among them, SALL4 is a C2H2 ZF-type neosubstrate for CRL4$^{CRBN}$ involved in embryonic limb development with a strong genetic link to embryopathies[30–33].

Thalidomide is mainly modified with 5-hydroxylation of the phthalimide moiety or 5′-hydroxylation of the glutarimide moiety through the action of cytochrome P450 isozymes[34–36]. Recently, it has been reported that 5-hydroxythalidomide (5HT) has distinct neosubstrate selectivity between IKZF1 and SALL4, and that 5HT induces degradation of SALL4, but not IKZF1[28]. Furthermore, 5HT induces more stronger degradation of SALL4 than thalidomide[28]. Therefore, it is predicted that 5HT contributes to thalidomide teratogenicity caused by SALL4 degradation. The molecular basis of its selectivity and degradation strength, however, has not been elucidated yet. The mechanistic insights into the C2H2 ZF-type neosubstrate selectivity of thalidomide metabolites will promote the understanding of the pharmaceutical actions of these IMiDs, which is required to develop thalidomide derivatives with reduced adverse effects by preventing unwanted off-target degradation. Here, we present structural bases for altered neosubstrate specificities of IMiDs through their metabolism. The crystal structure of the 5HT-mediated human SALL4–CRBN complex and mutagenesis studies elucidate that the complex formation is enhanced by the interaction between CRBN and an additional hydroxy group of 5HT. Our data also conclude that the variation in the second residue of β-hairpin

structure defines the C2H2 ZF-type neosubstrate selectivity of 5HT.

## Results

**Enantioselective 5HT action on the SALL4–CRBN interaction.** 5HT induces the interaction between human SALL4 and CRBN similarly to its precursor drug, thalidomide[28]. Although there are (S)- and (R)-enantiomers of thalidomide and its derivatives, the enantioselective action of thalidomide metabolites has not been investigated with regard to neosubstrate recruitment to CRBN. We therefore characterized the enantioselectivity of 5HT on the formation of the SALL4–CRBN complex compared with that of thalidomide using an AlphaScreen (AS)-based interaction assay (Fig. 1a). The (S)-enantiomer of 5HT was more effective in inducing complex formation than the (R)-enantiomer, and the same tendency was observed for thalidomide (Fig. 1b). The effective concentration of (R)-enantiomers was 10-fold higher than shown by the (S)-enantiomers, which is similar to the relative binding affinity of the thalidomide enantiomers for CRBN[37]. The results suggest that the 5-hydroxylation of the phthalimide moiety does not affect the enantioselectivity of thalidomide for the SALL4–CRBN complex. On the other hand, this modification enhances complex formation, as indicated by the AS signal being saturated at a lower concentration of 5HT than it was with thalidomide.

**Structure of the 5HT-mediated SALL4–CRBN complex.** To elucidate the SALL4–CRBN interaction mediated by the (S)-enantiomers of 5HT and thalidomide, we determined the high-resolution crystal structures of the ternary complexes (1.80 and 1.90 Å, respectively) by using the thalidomide-binding domain (TBD) of human CRBN (residues 318–426 in which C366 is mutated to Ser for the crystallization) and human SALL4 ZF2 (residues 410–432) (Fig. 1c). The mutation to Ser at residue 366 of the CRBN TBD, which is an evolutionarily conserved positive selection site[38], showed no influence on the enantioselectivity for SALL4–CRBN complex formation (Supplementary Fig. 1).

The CRBN TBD adopts a twisted β-sheet (β3–β4–β8–β7–β6–β5) anchoring a β-hairpin (β1 and β2) with a zinc ion (Zn$^{2+}$) that binds with four cysteine residues (C323, C326, C391, and C394) in two CXXC motifs (Fig. 1c and Supplementary Fig. 2). On the other hand, SALL4 ZF2 shows a typical structure of C2H2 ZF domains consisting of a β-hairpin (β1′ and β2′) and an α-helix (α1′), which are connected through Zn$^{2+}$ binding with the conserved CXXC and HXXXH motifs (C412, C415, H428, and H432) (Fig. 1c and Supplementary Fig. 2). SALL4 ZF2 contacts the open face of the twisted β-sheet of the CRBN TBD (Fig. 1c). Consistent with the results of the AS-based interaction assay, the C366S-mutated site is located far from the binding interface with SALL4 (Supplementary Fig. 2). Clear electron densities of 5HT and thalidomide were observed in the SALL4–CRBN complex, indicating the position of the 5-hydroxy group and the stereochemistry of the glutarimide ring in the (S)-forms (Fig. 1d). The (S)-enantiomers of 5HT and thalidomide are located on the interface between SALL4 ZF2 and the CRBN TBD with the same orientation (Fig. 1c).

The binding mode of SALL4 ZF2 is similar to that in the reported crystal structure of pomalidomide-mediated SALL4–CRBN–DDB1 complex[39], in which there is no contact of SALL4 ZF2 with regions other than the TBD (Fig. 2a). The position of β-hairpin loop relative to the CRBN TBD is completely aligned in the structures of three complexes mediated by 5HT, thalidomide, or pomalidomide. The W400 and H357 side chains of the CRBN TBD form direct hydrogen bonds with the backbone carbonyl groups on the β-hairpin loop of

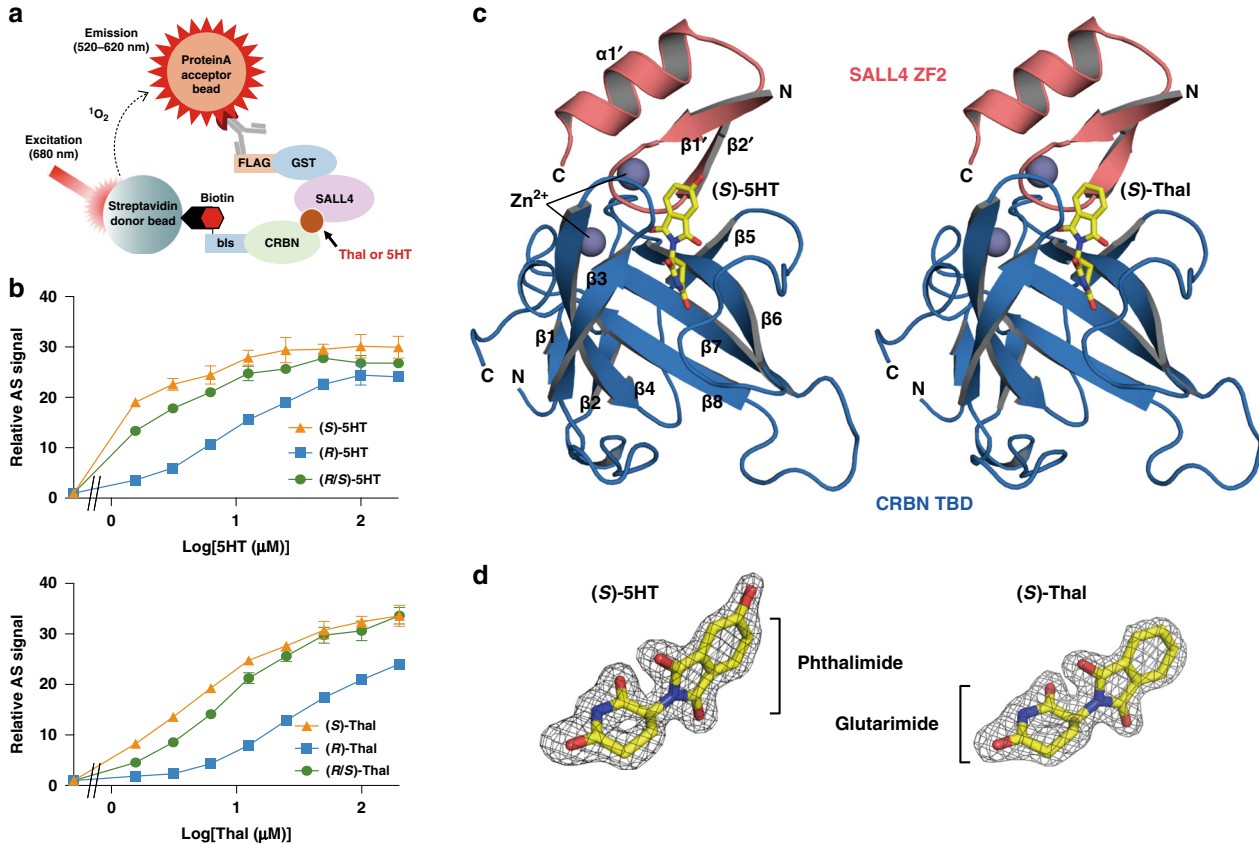

**Fig. 1 The formation of the SALL4 and CRBN complex as mediated by (S)-5HT or (S)-thalidomide (Thal). a** Schematic diagram of the AS-based interaction assay. **b** Dose-dependent formation of the SALL4–CRBN complex, as determined by the titration of racemate or each enantiomer of 5HT (upper) or Thal (lower) in the AS-based assay. AS signals are expressed as the luminescence signal relative to the luminescence signal of DMSO, which is considered equal to one. Data are presented as mean values ± standard deviation (SD) (n = 3 independent experiments). **c** Crystal structures of the ternary complex of CRBN TBD (blue) and SALL4 ZF2 (salmon) mediated by (S)-5HT (left) or (S)-Thal (right), which are shown by yellow sticks. Two zinc ions (Zn²⁺) are represented by the gray spheres in each complex. Secondary structural elements of CRBN TBD and SALL4 ZF2 are labeled in the (S)-5HT-mediated complex. **d** The structures of (S)-5HT (left) and (S)-Thal (right) in the SALL4–CRBN complex. The $F_o$–$F_c$ omit map is shown by a mesh diagram at a contour level of 4.0.

SALL4 ZF2 (V414, C415, and G416) (Fig. 2b). The formation of these hydrogen bonds is suggested in the structure of pomalidomide-bound CRBN complexed with SALL4 ZF2[39] and IKZF1 ZF2[24]. On the other hand, the Y355 side chain of the CRBN TBD, which is positioned near the β-hairpin loop of SALL4 ZF2, shows the different orientation from the pomalidomide-mediated complex (Fig. 2a). The residue is also located at the interface with the N-terminal domain (NTD) of CRBN, and the Y355A mutation decreases the pomalidomide-mediated SALL4–CRBN interaction[39]. Thus, the changes in the side-chain orientation may be due to the use of the truncated TBD and modulate the interaction between SALL4 ZF2 and the CRBN TBD. The other side-chain alteration was observed in the H378 residue of CRBN (Fig. 2a). In the complex structures, the H378 side chain is positioned in close proximity to the phthalimide moiety of (S)-5HT and (S)-thalidomide, but far from that of pomalidomide. Since the 4-amino group of pomalidomide directs to the H378 residue, the differences in its side-chain orientation may be due to the structural modification of these compounds.

The overall orientation of SALL4 ZF2 to the CRBN TBD is slightly different in the (S)-5HT- and (S)-thalidomide-mediated structures as compared with the pomalidomide-mediated structure (Fig. 2a), whereas this difference in orientation does not appear to affect the interaction between SALL4 ZF2 and the CRBN TBD. On the other hand, the improved resolution of the

complex structure in this study shows evidence of some hydrogen-bonding networks on the interface between SALL4 ZF2 and the CRBN TBD. The H417 and R418 side chains of SALL4 ZF2 form hydrogen bonds with the residues of the CRBN TBD through water molecules (Fig. 2c, d). However, these residues are not conserved in the C2H2 ZF domains, including IKZF1 ZF2 (Fig. 2e), and the H417 residue contributes to different orientation between SALL4 ZF2 and IKZF1 ZF2 toward CRBN TBD[39]. Besides, in the (S)-thalidomide-mediated complex, there is no hydrogen-bonding network involving R418 (Fig. 2d). The contribution of R418 for the SALL4–CRBN interaction may be related to the change in the polar environment of the interface between SALL4 and CRBN, which is caused by the substitution of the 5-hydroxy group at the phthalimide moiety of thalidomide.

**Binding mode of the (S)-enantiomer of 5HT.** The backbone structure of (S)-5HT completely overlaps with (S)-thalidomide in the SALL4–CRBN complex (Fig. 2a and Supplementary Fig. 3). The glutarimide moiety is enveloped in the twisted β-sheet of the CRBN TBD and forms van der Waals contacts with three Trp residues (W380, W386, and W400) (Fig. 3a and Supplementary Fig. 4). This structural evidence confirms that 5-hydroxylation has no effect on the enantioselectivity of the SALL4–CRBN interaction. The preferred (S)-enantioselectivity of CRBN has been explained by the lower conformational energy lost upon a

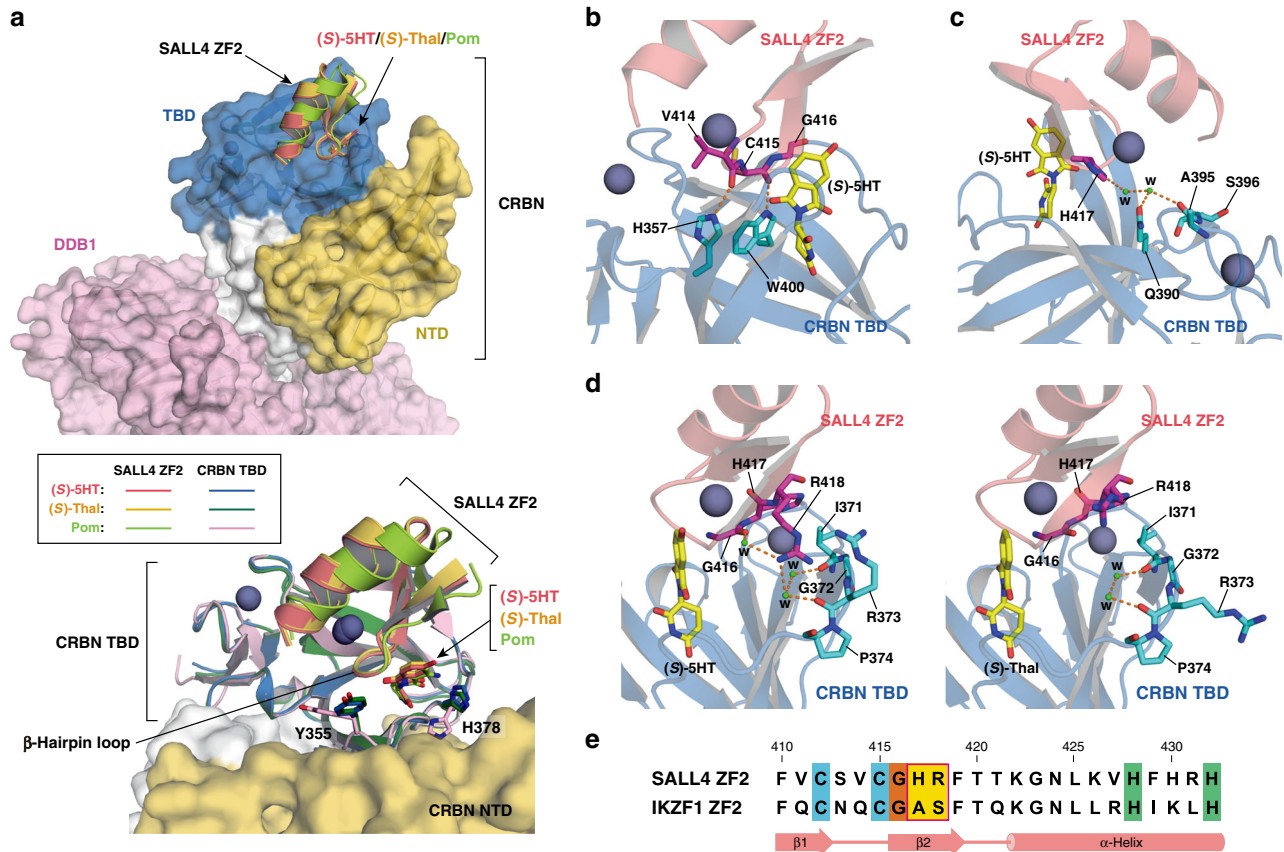

**Fig. 2 Interactions between SALL4 ZF2 and CRBN TBD. a** Structural comparison among the ternary complexes mediated by (*S*)-5HT, (*S*)-thalidomide (Thal), and pomalidomide (Pom). In the upper panel, a CRBN molecule is separated into three regions, N-terminal domain (NTD, yellow-orange), TBD (blue), and the region connecting NTD and TBD (white), which provides the interface for DDB1 (pink). In the lower panel, CRBN TBD is shown by ribbon diagrams as well as SALL4 ZF2. The colored lines in the box represent color arrangement of SALL4 ZF2 and CRBN TBD in the three complex states mediated by (*S*)-5HT, (*S*)-Thal, and Pom. The structure of the SALL4 ZF2–CRBN–DDB1 complex with Pom was generated from the coordinate data deposited in the PDB under accession number 6UML[39]. **b** Hydrogen bonds formed between the β-hairpin loop of SALL4 ZF2 and residues H357 and W400 of CRBN. **c** Water-mediated hydrogen-bonding network between SALL4 H417 and the residues of the CRBN TBD. **d** Different orientations and hydrogen-bonding networks through water molecules of SALL4 R418 in complex with the CRBN TBD as mediated by (*S*)-5HT (left) or (*S*)-thalidomide (Thal) (right). The hydrogen bonds and water molecules are represented by orange dashed lines and dots labeled w, respectively. Gray spheres are zinc ions bound to CRBN TBD and SALL4 ZF2. **e** Sequence alignment of SALL4 ZF2 and IKZF1 ZF2. Residues conserved in C2H2 ZF domains are shaded in cyan (Cys), orange (Gly), or lime (His). The residues illustrated in **c**, **d** are shaded yellow in magenta box. Numbers correspond to SALL4 ZF2.

shift in the phthalimide moiety towards the *endo* direction in the (*S*)-form than is realized in the (*R*)-form, which is based on the binding mode of (*S*)- or (*R*)-thalidomide to the neosubstrate-free state of the mouse CRBN TBD (97.2% sequence identity to the human CRBN TBD)[37]. In the (*S*)-thalidomide-bound SALL4–CRBN complex, the phthalimide moiety moves farther from the neosubstrate-free state towards the *endo* direction to avoid steric hindrance caused by the β-hairpin loop in SALL4 ZF2 and the β3–β4 loop in the CRBN TBD, conformation of which seems to be fixed upon contact with SALL4 ZF2 (Fig. 3b, c). Therefore, the favorable SALL4–CRBN interaction mediated by the (*S*)-enantiomer of 5HT or thalidomide can be defined by its binding preference for SALL4 and CRBN.

The phthalimide moiety with the 5-hydroxy group of (*S*)-5HT is positioned between SALL4 ZF2 and the CRBN TBD as a molecular glue that enhances the protein–protein interaction (Fig. 3a). There are several van der Waals contacts and hydrophobic interactions of the phthalimide moiety with the CRBN TBD and SALL4 ZF2. In addition to residue N351 of the CRBN TBD, a hydrogen bond is also formed between the 5-hydroxy group of (*S*)-5HT and the H353 side chain through a water molecule, which contributes to fix the orientation of the

phthalimide moiety modified with the 5-hydroxy group (Fig. 3a and Supplementary Fig. 4). On the other hand, (*S*)-5HT has no hydrogen bond with SALL4 ZF2. Isothermal titration calorimetry (ITC) experiments were used to evaluate the binding affinity of (*S*)-5HT toward the CRBN TBD and its H353A mutant. The data showed that (*S*)-5HT bound to the CRBN TBD with a $K_D$ value of $0.76 \pm 0.20\,\mu M$, and this affinity was decreased by an H353A mutation ($2.28 \pm 0.10\,\mu M$) (Fig. 3d). In contrast, the $K_D$ values of (*S*)-thalidomide were almost the same toward the wild type ($4.00 \pm 0.36\,\mu M$) and H353A mutant ($4.43 \pm 0.16\,\mu M$) of the CRBN TBD. These results indicate that (*S*)-5HT has a higher binding affinity toward CRBN TBD than (*S*)-thalidomide with the major contribution of the additional hydrogen-bond formation with the H353 side chain. The enhanced binding of (*S*)-5HT toward CRBN TBD partially explains the more profound 5HT-mediated formation of the SALL4–CRBN complex than that mediated by thalidomide.

We further examined the effect of the H353A mutation on the SALL4–CRBN interaction mediated by (*S*)-5HT using an AS-based assay. The interaction was induced at a lower concentration of (*S*)-5HT than (*S*)-thalidomide, whereas the effective concentration of (*S*)-5HT was increased in the CRBN H353A mutant

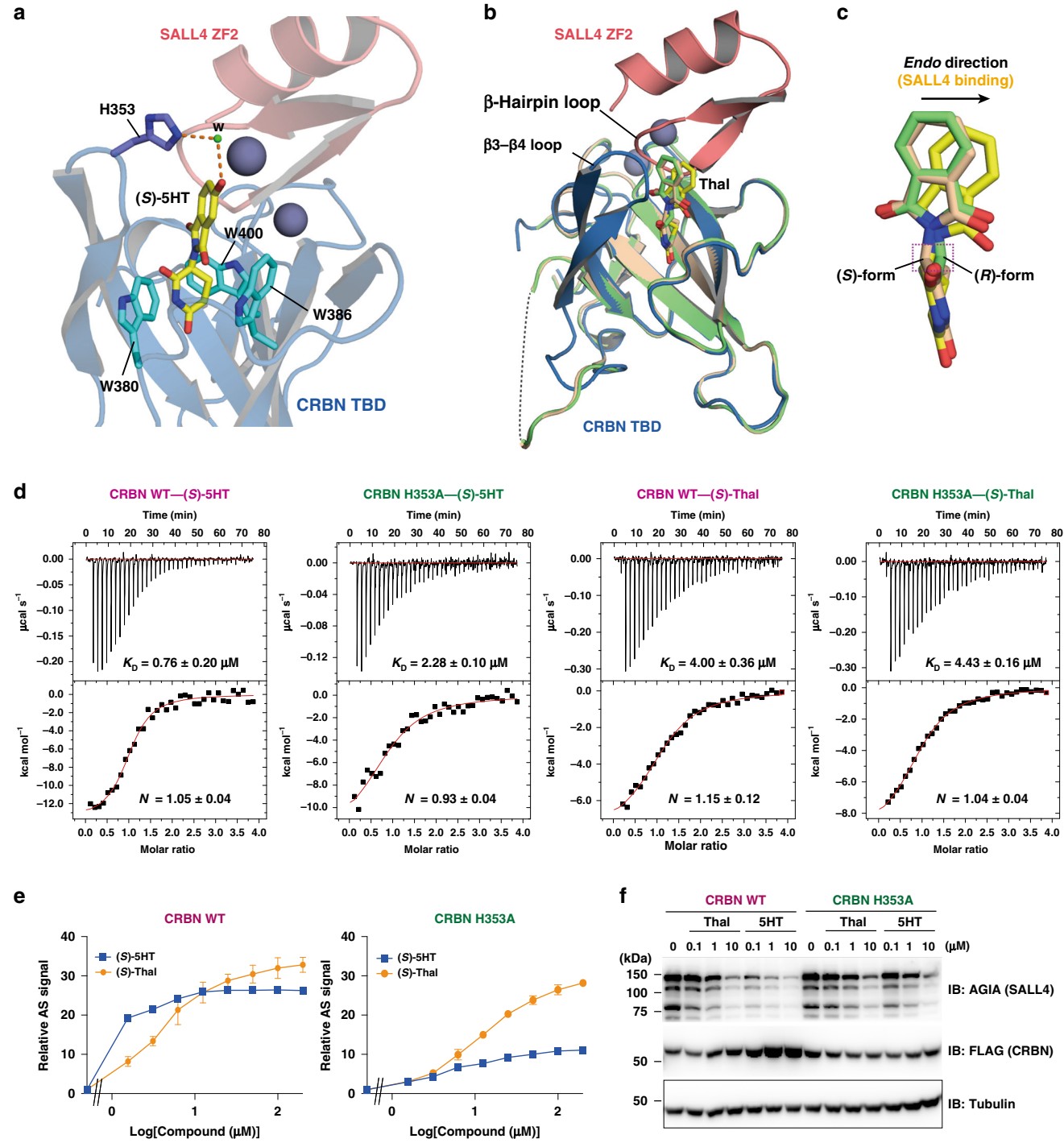

**Fig. 3 Structural bases for the effective (S)-5HT mediation of the CRBN–SALL4 interaction. a** The binding mode of (S)-5HT on the interface between the CRBN TBD and SALL4 ZF2. The H353 residue forms hydrogen bonds (dashed lines) with the 5-hydroxy group of (S)-5HT through a water molecule (dot labeled w). Gray spheres are zinc ions bound to the CRBN TBD and SALL4 ZF2. **b** Superimposed structures of the CRBN TBD in complex with SALL4 and (S)-thalidomide (Thal, blue), neosubstrate-free state binding of (S)-Thal (wheat) and (R)-Thal (lime). (S)-Thal in the SALL4–CRBN complex is represented by yellow sticks, and the stick models of (S)- and (R)-Thal are colored the same as the coordinates in CRBN in the neosubstrate-free state. Zinc ions are shown as gray spheres. The black dashed line connects the missing chain of the neosubstrate-free CRBN TBD, which corresponds to a part of two β-strands, β3 and β4, observed in the SALL4–CRBN complex. The structures of the neosubstrate-free CRBN with (S)- and (R)-Thal were generated from the coordinates deposited in the PDB under accession numbers 5YJ0 and 5YJ1[37]. **c** Enlarged views of the stick models of Thal in **b**. **d** ITC thermograms of the wild-type (WT) and H353A mutant of the CRBN TBD titrated with (S)-5HT and (S)-Thal. The values of the dissociation constant ($K_D$) and molar binding ratio (N) were calculated with mean values ± SD (n = 3 independent experiments). **e** Effect of the H353A mutation in CRBN on the SALL4 interactions induced by (S)-5HT or (S)-Thal. AS signals are expressed as the relative luminescence signal relative to the luminescence signal of DMSO, which is considered equal to one. Data are presented as mean values ± SD (n = 3 independent experiments). **f** Effect of the H353A mutation in CRBN on the (S)-5HT- or (S)-Thal-mediated proteasomal degradation of SALL4. The experiment was repeated three times independently with similar results.

and was similar to that of (S)-thalidomide with a weakened AS signal (Fig. 3e and Supplementary Fig. 5). In addition, the effect of (S)-thalidomide on the SALL4–CRBN interaction was not largely changed by the H353A mutation. The extent of the dose dependency for the proteasomal degradation of SALL4 was observed to be similar in the cell assays (Fig. 3f). Mediated by the CRBN H353A mutant, (S)-5HT induced the SALL4 degradation in the same concentration range as (S)-thalidomide did by both wild-type and mutated CRBN. These compatible data led to the following conclusions: (i) the enhanced formation of the SALL4–CRBN complex requires a hydrogen bond between the 5-hydroxy group of (S)-5HT and the H353 residue in CRBN, and (ii) the thalidomide-mediated proteasomal degradation of SALL4 is principally regulated by the interaction of ZF2 with the CRBN TBD.

**Key structural basis for the neosubstrate selectivity of 5HT.** In a previous study, 5HT did not mediate an interaction between CRBN and IKZF1, which is another C2H2 ZF-type neosubstrate[28]. To determine the structural basis for the neosubstrate selectivity of 5HT, we compared the structures of SALL4 ZF2 and IKZF1 ZF2[24] in complex with the CRBN TBD as mediated by (S)-5HT and a thalidomide derivative, pomalidomide, respectively. Upon binding to IKZF1 ZF2, pomalidomide spatially overlapped with (S)-5HT and (S)-thalidomide in complex with SALL4 ZF2, except for the substituted groups on the phthalimide moiety, as shown when the structure of the CRBN TBD was superimposed (Supplementary Fig. 6). The β-hairpin loops in SALL4 ZF2 and IKZF1 ZF2 are also located at the same position relative to the CRBN TBD and the thalidomide derivatives, although the overall orientation of the ZF2 in each was relatively different. The superimposed structures indicated differences in two ZF2 residues located around the 5-hydroxy group of the (S)-5HT and related to the distinct action of 5HT on neosubstrates: V411 and R418 in SALL4 ZF2 that are the spatially and sequentially corresponding residues of Q146 and S153 in IKZF1 ZF2, respectively (Fig. 4a). These residues are positioned as second and ninth resides at the β-hairpin structure of C2H2 ZF domain (Fig. 4b).

To identify the residues controlling the C2H2 ZF-type neosubstrate selectivity of 5HT, the interaction of CRBN with SALL4 and IKZF1 was evaluated by using the residue-swap ZF2 mutants. The AS-based assay showed that the (S)-5HT-mediated SALL4–CRBN interaction was dramatically decreased by the mutation of the V411 residue to a Gln residue of IKZF1 (Fig. 4c). The residue-swap effect was not enhanced by the addition of the IKZF1-mimic mutation at R418 (SALL4 V411Q/R418S double mutant). In contrast, these mutations showed no influence on the (S)-thalidomide-mediated SALL4–CRBN interaction (Fig. 4c). The effects of the residue swap in the SALL4 mutants were consistent with the results observed in the proteasomal degradation assays (Fig. 4d). The SALL4 V411Q and V411Q/R418S mutants were not degraded by (S)-5HT, even when its concentration was increased to equal the saturation concentration for the SALL4–CRBN interaction. These results conclude that V411 residue of SALL4 is the critical residue for the effective activity of (S)-5HT in the formation of the SALL4–CRBN complex. The V411Q mutation may cause steric hindrance with the 5-hydroxy group of (S)-5HT in the SALL4–CRBN complex because the Gln residue has a bulkier side chain than the Val residue (Fig. 4a).

The SALL4-mimic mutation of IKZF1 was also evaluated by using the AS-based assay. The (S)-5HT-mediated IKZF1–CRBN interaction was detected by the residue-swap Q146V mutation in IKZF1, and the effect was slightly increased by the addition of the

S153R mutation (IKZF1 Q146V/S153R double mutant) (Fig. 4e). These results support that the variation in the second residue of β-hairpin structure mainly contributes to the C2H2 ZF-type neosubstrate selectivity of 5HT. However, in contrast to SALL4, the mutations of IKZF1 weaken the (S)-thalidomide-mediated IKZF1–CRBN interaction. The proteasomal degradation of IKZF1 showed a dose dependency similar to the profiles from the AS-based interaction assay in each residue-swap mutant (Fig. 4f). These data indicate that residues Q146 and S153 cause IKZF1 to adapt to interact with CRBN as mediated by (S)-thalidomide, but not by (S)-5HT.

The V411 residue of SALL4, which corresponds to the second residue of β-hairpin structure, is required for the effect of (S)-5HT on its interaction with CRBN. The C2H2 ZF domains capable of binding to CRBN in a pomalidomide-dependent manner[24] have adopted several residue types in the same position as V411 in SALL4: Ala, Thr, Val (SALL4 type), Ile, Leu, His, Gln (IKZF1 type), Glu, Lys, and Arg (Fig. 5a). The 5-hydroxylation of pomalidomide impairs the degradation of IKZF1, but not SALL4, through the interaction with CRBN as well as (S)-5HT (Fig. 5b–d). We evaluated the possibility that these CRBN neosubstrates may exhibit different (S)-5HT or 5-hydroxypomalidomide (5HP) dependencies by the AS-based interaction assay of SALL4 with a V411 mutation. The V411T and V411I mutants induced the complex formation with CRBN at the same concentration range of (S)-5HT as the wild-type SALL4, and the V411A mutant also formed the complex with a weakened AS signal (Fig. 5e). In contrast, (S)-5HT-mediated interaction with CRBN was largely impaired in the SALL4 mutants with a side chain larger than Leu residue, including the IKZF1 type. Although the V411 mutation of SALL4 also affected the (S)-thalidomide-induced interaction, the V411L, V411Q, and V411E mutants retained the binding ability to CRBN as well as the wild-type SALL4. Similarly, the influence of the variation in the second residue of β-hairpin structure was also observed in the SALL4–CRBN interaction mediated by 5HP or pomalidomide (Fig. 5e). These data suggest that 5-hydroxylation modification of phthalimide moiety tightens the C2H2 ZF-type neosubstrate selectivity of thalidomide and its derivatives.

## Discussion

In recent studies, SALL4, PLZF, and p63 have been reported to be the neosubstrates involved in thalidomide teratogenicity[26–29]. Therefore, it is thought that several neosubstrates play functional roles in thalidomide teratogenicity. SALL4 is one of the most potential candidates involved in thalidomide teratogenicity due to both genetic evidence of SALL4[30–33] and sensitivity of SALL4 degradation to thalidomide[26–28]. Furthermore, 5HT induced degradation of both SALL4 and PLZF, but not IKZF1 (Fig. 4c–f)[28], and 5HT induces SALL4 degradation more strongly than thalidomide (Fig. 3f). Recently, we showed that 5HT induced similar teratogenic phenotypes to thalidomide in chicken limb bud[28]. In this study, we conclusively demonstrated the structural bases for the (S)-5HT-mediated formation of the SALL4–CRBN complex and its selectivity for C2H2 ZF-type neosubstrates, providing structural evidence that the glue-type E3 ligase modulators cause altered neosubstrate specificities through metabolism.

Thalidomide and its derivatives are typical drugs, which show species specificity, and it has been reported that thalidomide does not show its teratogenic phenotypes in mice[40]. Actually, the degradation of neosubstrates is mediated by CRBNs derived from highly sensitive species to thalidomide, including human and rabbit, but does not occur through mice CRBN[19,26–28]. In addition, thalidomide low-sensitivity species, such as chicken, show the degradation of PLZF but not SALL4 by a thalidomide

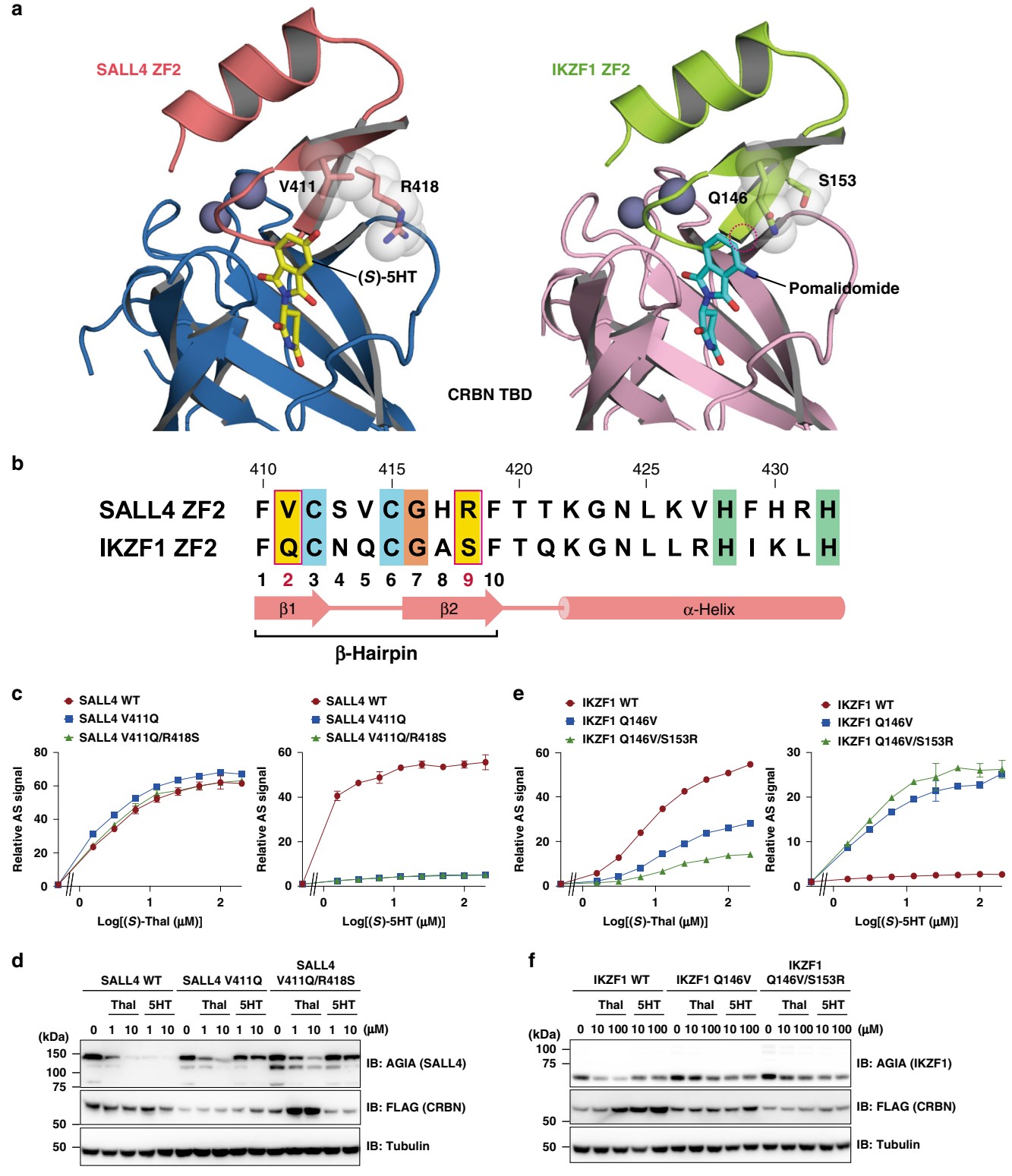

treatment[28]. Currently, it is thought that these differences of
sensitivity to thalidomide result from differences of amino acid
sequences of both CRBN and neosubstrates between the
species[19,26–28]. Therefore, these evidences and the molecular basis
of 5HT-mediated neosubstrate degradation of CRBN strongly
suggest that degradation of SALL4, PLZF, and other proteins by
both thalidomide and 5HT causes severe teratogenic phenotypes
in highly sensitive species to thalidomide. However, many further
researches using the highly sensitive species will be required to

show whether what range and what tissues are damaged by
thalidomide and 5HT through the molecular interactions pro-
posed in this study.

Pomalidomide and lenalidomide are effective IMiDs for the
treatment of multiple myeloma[7]. However, they also induce the
proteasomal degradation of SALL4 through its interaction with
CRBN, which causes adverse effects, including teratogenicity[28].
Furthermore, these thalidomide derivatives can be modified by
5-hydroxylation[41–43], and the 5-hydroxylated metabolites act

**Fig. 4 C2H2 ZF-type neosubstrate selectivity of (S)-5HT. a** Distinct residues located near the 5-hydroxy group of (S)-5HT in the ZF2 of SALL4 (V411 and R418) and IKZF1 (Q146 and S153). (S)-5HT and pomalidomide are shown by yellow and cyan sticks, respectively. Gray spheres are zinc ions observed in both complex structures. The magenta dashed-line circle indicates the position of the 5-hydroxy group on the phthalimide moiety. The structure of the IKZF1–CRBN complex with pomalidomide was generated from the coordinate data deposited in the PDB under accession number 6HOF[24]. **b** Sequence alignment of SALL4 ZF2 and IKZF1 ZF2. Residues conserved in C2H2 ZF domains are shaded in cyan (Cys), orange (Gly), or lime (His). The residues illustrated in **a** are shaded yellow in magenta boxes. Numbers on the sequences correspond to SALL4 ZF2. The residue position of β-hairpin structure is indicated on the diagram of secondary structure. **c**, **e** Effect of the residue-swap mutations in SALL4 (**c**) and IKZF1 (**e**) on the CRBN interaction induced by (S)-5HT or (S)-thalidomide (Thal). AS signals are expressed as the relative luminescence signal relative to the luminescence signal of DMSO, which is considered equal to one. Data are presented as mean values ± SD (n = 3 independent experiments). **d**, **f** Effect of the residue-swap mutations in SALL4 (**d**) or IKZF1 (**f**) on the proteasomal degradation of each through their interaction with CRBN as mediated by (S)-5HT or (S)-Thal. The experiment was repeated three times independently with similar results.

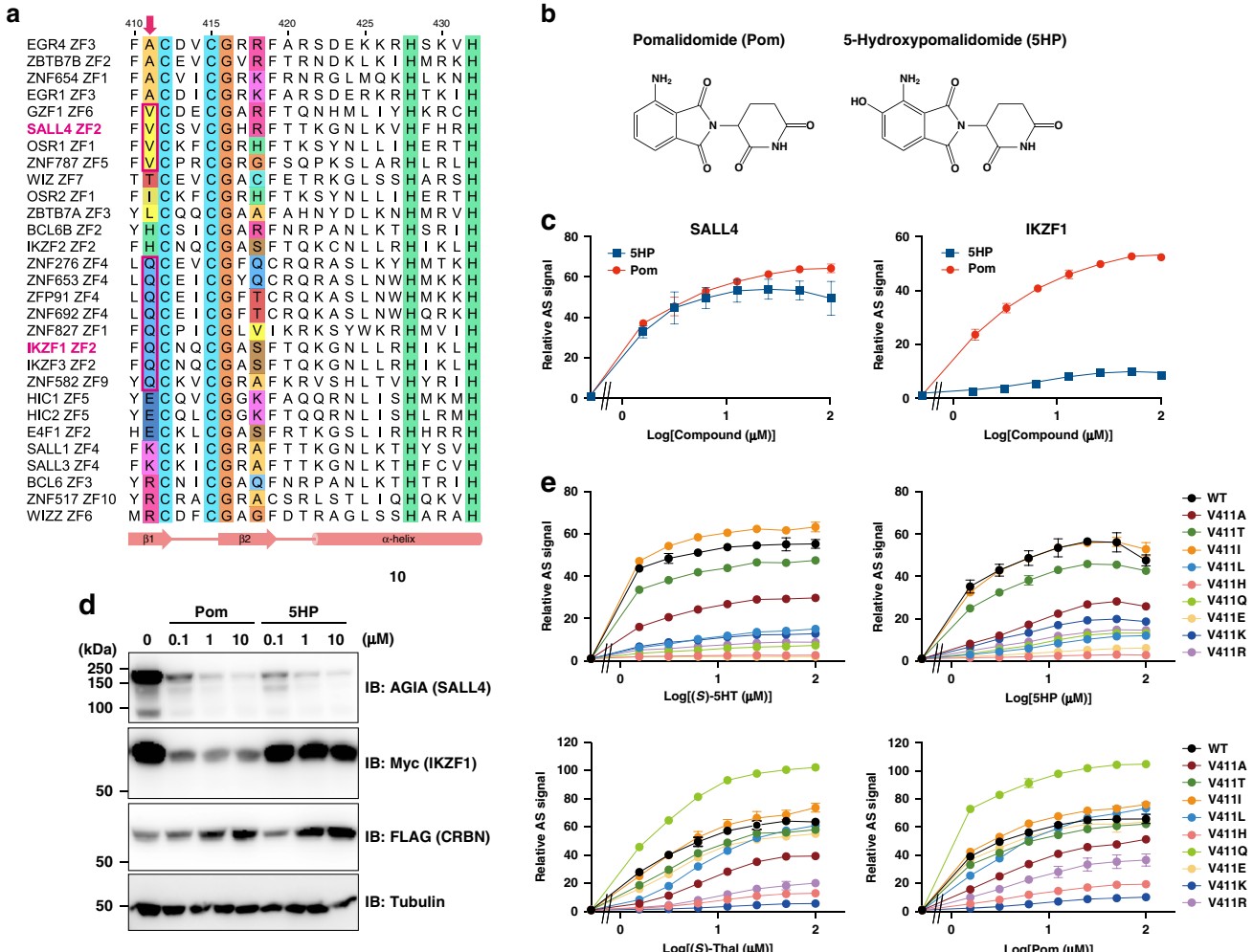

**Fig. 5 CRBN-binding selectivity is affected by the V411 residue in SALL4. a** Sequence alignment of SALL4 ZF2 and other C2H2 ZF domains according to the binding capacity of pomalidomide-engaged CRBN[24]. Residues conserved in the C2H2 ZF domains are shaded in cyan (Cys), orange (Gly), or lime (His). Arrow indicates the second residue of β-hairpin structure in C2H2 ZF-type neosubstrates. Numbers correspond to SALL4 ZF2. **b** Chemical structures of pomalidomide (Pom) and 5-hydroxypomalidomide (5HP). **c** Dose-dependent interaction of CRBN with SALL4 (left) and IKZF1 (right), as observed with the titration of racemate, Pom, and 5HP in the AS-based assay. AS signals are expressed as the luminescence signal relative to the luminescence signal of DMSO, which is considered equal to one. Data are presented as mean values ± SD (n = 3 independent experiments). **d** Comparison of Pom- or 5HP-dependent degradation of SALL4 and IKZF1. The experiment was repeated three times independently with similar results. **e** The effect of V411 mutation in SALL4 on the CRBN interaction mediated by (S)-thalidomide (Thal), (S)-5HT, Pom, or 5HP. AS signals are expressed as the luminescence signal relative to the luminescence signal of DMSO, which is considered equal to one. Data are presented as mean values ± SD (n = 3 independent experiments).

selectively on SALL4 and not on IKZF1 (Fig. 5c, d), the degradation of which is linked to the anti-proliferative and immunomodulatory effects of IMiDs[16–18]. Therefore, avoidance of the impact of 5-hydroxylation modification on the phthalimide moiety of IMiDs may reduce thalidomide teratogenicity affected

by SALL4 degradation. One of the approaches is the design of thalidomide derivatives based on the C2H2 ZF-type neosubstrate selectivity that depends on the 5-hydroxylation modification of the phthalimide moiety. The structural bases available for use in this approach are characterized by differences in two ZF2 residues

located around the 5-hydroxy groups in 5HT, for example, V411 and R418 in SALL4 (Fig. 4a). In this study, the major interaction between SALL4 ZF2 and CRBN can be observed by using the truncated TBD as a good proxy for the full-length CRBN, whereas some neosubstrates for PROTACs, such as BRD4[44] and GSPT1[23], interact not only with the TBD but also with the NTD of CRBN. These structural evidences suggest that the use of the truncated TBD is restricted by an interaction mode of neosubstrate with CRBN. At least the interaction of a single C2H2 ZF domain, including SALL4 ZF2 and IKZF1 ZF2, can be analyzed by using the truncated TBD, which are helpful for the structure-guided drug design by enabling the visualization of the atomic structures of candidate compounds bound to the CRBN–C2H2 ZF complex. Thus, our results provide an emerging conceptual framework for more effective design of IMiDs with reduced off-target degradation and few adverse effects by selectively degrading C2H2 ZF-containing proteins through CRL4$^{CRBN}$.

## Methods

**Reagents and antibodies.** (S/R)-thalidomide (Sigma-Aldrich and Tokyo Chemical Industry Co., Ltd.), (S)-thalidomide (Sigma-Aldrich), (R)-thalidomide (Sigma-Aldrich), pomalidomide (Sigma-Aldrich), lenalidomide (FUJIFILM Wako Pure Chemical Corporation), (S/R)-5HT were prepared according to a previously published method[45]. (S)-5HT, (R)-5HT, and (S/R)-5HT (Enamine) at 50 mM each were dissolved in dimethyl sulfoxide (DMSO; FUJIFILM Wako Pure Chemical Corporation) and stored at −20 °C as stock solutions. All the drugs were diluted 1000-fold for the cell degradation analyses or diluted 200-fold for the AS-based analyses using the AS technology. For the crystallization and ITC experiments, (S)-thalidomide and (S)-5-hydroxythalidomide at 1 M were dissolved in DMSO as stock solutions.

In this study, the following antibodies were used at each dilution ratio: anti-FLAG mouse monoclonal antibody (mAb) (1:5000, horseradish peroxidase (HRP)-conjugated, Sigma-Aldrich, A8592), anti-AGIA rabbit mAb[46] (1:10,000, HRP-conjugated, produced in our laboratory), and anti-Myc mouse mAb (1:3000, HRP-conjugated, Cell Signaling Technology, 2276) were used for the detection of epitope-tagged proteins. Anti-α-tubulin rabbit polyclonal Ab (1:10,000, HRP-conjugated, MBL, PM054-7) was used to detect α-tubulin. Biotinylated proteins were detected by anti-biotin (1:5000, HRP-conjugated, Cell Signaling Technology, 7075).

**Plasmids for the AS-based and cell degradation assays.** pDONR221 and pcDNA3.1(+) plasmids, based on Gateway technology, were purchased from Invitrogen, and the pEU vector for the wheat cell-free system was constructed in our laboratory[47]. pcDNA3.1(+)-FLAG-GW, pcDNA3.1(+)-AGIA-MCS, pcDNA3.1(+)-Myc-MCS, and pEU-bls-GW plasmids were constructed based on each original vector by using the In-Fusion system (TaKaRa Bio) or restriction enzymes. pEU-FLAG-GST-IKZF1 and pEU-FLAG-GST-SALL4 were purchased from the Kazusa DNA Research Institute. Open reading frames (ORFs) in SALL4 and IKZF1 were amplified and restriction enzyme sites were added by PCR and cloned into pcDNA3.1(+)-AGIA-MCS or pcDNA3.1(+)-Myc-MCS. The ORF of CRBN was purchased from the Mammalian Gene Collection[48]. CRBN was amplified, and the BP reaction sequence (attB and attP) was added by PCR and cloned into pDONR221 using BP recombination (Invitrogen). Then, pDONR221-CRBN was recombined into pEU-bls-GW or pcDNA3.1(+)-FLAG-GW using LR recombination (attL and attR). Amino acid mutation in each protein was generated by inverse PCR and In-Fusion.

**Protein expression and purification.** DNA sequences encoded human CRBN TBD (318–426 and C366S mutation) and SALL4 ZF2 (410–432) were cloned into pGEX6P-3 (GE Healthcare), and the recombinant proteins were expressed in *Escherichia coli* Rossetta(DE3) (Novagen) using lysogeny-broth media supplemented with 20 μM ZnCl$_2$, 100 μg ml$^{-1}$ ampicillin, and 17 μg ml$^{-1}$ chloramphenicol. Protein expression was induced by adding 0.5 mM isopropyl β-D-thiogalactopyranoside at 18 °C when the optical density at 600 nm (OD$_{600}$) reached ~0.6. The cells were collected by centrifugation and were then resuspended in buffer containing 20 mM Tris-HCl, pH 8.0, 500 mM NaCl, and 0.1 mM tris(2-carboxyethyl)phosphine (TCEP). After sonication and centrifugation, the supernatant of the cell lysate was passed over glutathione Sepharose 4B resin (GE Healthcare), and resin-bound protein was cleaved overnight by human rhinovirus 3C protease. Proteins eluted from the resin were purified by size-exclusion chromatography with Superdex 75 10/300 GL (GE Healthcare) in 50 mM HEPES-NaOH, pH 7.4, 200 mM NaCl, and 0.1 mM TCEP. The fractions containing the CRBN TBD or SALL4 ZF2 were pooled and concentrated by ultrafiltration with Vivaspin 20 (MWCO 3000, Sartorius) to ~750 μM for the CRBN TBD and 100–150 μM for SALL4 ZF2. The proteins were stored at −80 °C. The concentration of

the CRBN TBD was determined by measuring the absorbance at 280 nm, and the molecular extinction coefficient was 27,960 M$^{-1}$ cm$^{-1}$ using Bradford protein assay kit (Thermo Fisher Scientific) was used to measure the concentration of SALL4 ZF2, with bovine serum albumin (BSA) serving as the protein standard.

For the AS-based interaction assay, recombinant SALL4–CRBN and IKZF1 were synthesized using a wheat cell-free system. In vitro transcription and translation based on wheat cell-free protein synthesis were performed using a WEPRO1240 expression kit (Cell-Free Sciences). Transcription reactions were conducted by SP6 RNA polymerase using DNA fragments as templates. The translation reactions were performed by the bilayer method using a WEPRO1240 expression kit according to the manufacturer's protocol. For the synthesis of biotinylated CRBN, 1 μl of cell-free synthesized crude biotin ligase (BirA) was added to the bottom layer, and 0.5 μM (final concentration) of D-biotin (Nacalai Tesque) was added to both the top and bottom layers[49].

**Crystallization and data collection.** The CRBN TBD and SALL4 ZF2 complex was crystallized by sitting-drop vapor diffusion in the presence of (S)-thalidomide or (S)-5HT. The SALL4–CRBN complex solution (104 μM) was prepared by mixing SALL4 ZF2 and the CRBN TBD with 2 mM (S)-thalidomide. The solution was mixed with an equal volume of a reservoir solution containing 27% (w/v) polyethylene glycol (PEG) 4000, 0.1 M sodium acetate, pH 5.5, and 0.1 M MgCl$_2$, which was then equilibrated against the reservoir solution at 20 °C. For the (S)-5HT-bound SALL4–CRBN complex (70 μM), SALL4 ZF2 was mixed with the CRBN TBD and 1.8 mM (S)-5HT. The complex solution was further mixed with an equal volume of a reservoir solution containing 22% (w/v) PEG 4000, 0.1 M MES-NaOH, pH 6.0, and 0.2 M Li$_2$SO$_4$, which was then equilibrated against the reservoir solution at 20 °C.

Each obtained crystal was soaked in the reservoir solution containing 25% (v/v) ethylene glycol as a cryoprotectant using dual thickness MicroMounts (MiTeGen) and then cooled in a liquid nitrogen stream. Diffraction data were collected using a Pilatus 2M-F with an AR-NE3A beamline at the Photo Factory (Tsukuba, Japan). The data were indexed and integrated using XDS[50] and scaled using AIMLESS[51]. The crystals of the SALL4–CRBN complex belonged to the space group C222$_1$ with unit cell parameters of $a$ = 70.96, $b$ = 92.72, and $c$ = 43.99 Å for (S)-thalidomide, and $a$ = 83.62, $b$ = 93.89 and $c$ = 43.68 Å for (S)-5HT. The data collection statistics are summarized in Supplementary Table 1.

**Structure determination and refinement.** The ternary structures of SALL4 ZF2, CRBN TBD, and (S)-thalidomide were determined by molecular replacement with Phaser-MR of the PHENIX program suite[52] using human CRBN TBD (Protein Data Bank (PDB) code: 4TZ4) and IKZF1 (PDB code: 6H0F) as search models. The ternary structures of the CRBN TBD, SALL4 ZF2, and (S)-5HT were determined by molecular replacement in the SALL4–CRBN complex with (S)-thalidomide, as determined in the present study. The iterative model building and refinement cycles were performed using COOT[53] and phenix.refine[54]. All structures were generated with PyMOL (Schrödinger), and the model quality was evaluated with MolProbity[55]. The refinement statistics are summarized in Supplementary Table 1.

**ITC measurements.** The binding affinity of (S)-5HT to the CRBN TBD was measured by using an isothermal titration calorimeter (MicroCal iTC$_{200}$, Malvern) with a reference power of 5 μcal s$^{-1}$ and stirring speed of 750 r.p.m. at 25 °C. The CRBN TBD (wild-type or H353A mutant) was dialyzed in a binding buffer containing 50 mM sodium phosphate, pH 7.4, 200 mM NaCl, and 0.1 mM TCEP, and then DMSO was added to the protein solution at a final concentration of 0.2%. (S)-5HT or (S)-thalidomide was dissolved in DMSO, and the solution was mixed with binding buffer with the DMSO concentration adjusted to 0.2%. For titrations, the (S)-5HT solution (200 μM) or (S)-thalidomide (400 μM) was injected into the sample cell filled with the CRBN TBD solution (10 or 20 μM) in 37 consecutive 1.0-μl aliquots at 120-s intervals. The first injection volume was 0.4 μl, and the observed thermal peak was excluded from the data analyses. Data fitting was performed using the Origin 7.0 software (OriginLab) in the one set of sites mode. The values of the dissociation constant ($K_D$) and molar binding ratio ($N$) were calculated with mean values ± SD ($n$ = 3 independent experiments).

**AS-based interaction assays.** For this assay, we directly used translational mixtures from the wheat cell-free protein production system. Then, 0.5 μl of biotinylated CRBN and 0.8 μl of FLAG-GST-SALL4 or FLAG-GST-IKZF1 was mixed in 15 μl of AS buffer containing 100 mM Tris (pH 8.0), 0.01% Tween-20, 100 mM NaCl, and 1 mg ml$^{-1}$ BSA. Then, 5 μl of a mixture containing 0.0125 μl of IMiD in AS buffer was added, and 20 μl of the mixture was incubated at 26 °C for 1 h in a 384-well AlphaPlate (PerkinElmer). Next, 5 μl of detection mixture containing 0.2 μg ml$^{-1}$ anti-DYKDDDDK mouse mAb (dilution ratio of 1:2500, FUJIFILM Wako Pure Chemical Corporation), 0.08 μl of streptavidin-coated donor beads, and 0.08 μl of protein A-coated acceptor beads (PerkinElmer) in AS buffer was added to each well. After incubation at 26 °C for 1 h, luminescence signals were measured using an Envision plate reader (PerkinElmer).

**Cell culture and transfection.** HEK293T cells were cultured in Dulbecco's modified Eagle's medium (low glucose) (FUJIFILM Wako Pure Chemical Corporation)

supplemented with 10% fetal bovine serum (FUJIFILM Wako Pure Chemical Corporation), 100 U ml$^{-1}$ penicillin, and 100 µg ml$^{-1}$ streptomycin (Gibco) at 37 °C and 5% CO$_2$. HEK293T cells were transfected using PEI Max: poly-ethyleneimine "Max" (MW 40,000) (PolyScience, Inc.).

**Generation of *CRBN*-knockout (KO) HEK293T cells**. For the generation of *CRBN*-KO HEK293T cells, the guide nucleotide sequence 5′-ACTCCGGGCG GTTACCAGGC-3′ was selected from the human *CRBN* gene. The *CRBN*-KO HEK293T cells were generated as previously published method[56] by CRISPR/Cas9-mediated genome editing.

**IMiD-induced proteasomal degradation assay**. For experiments using mutant CRBN, the HEK293T-*CRBN*$^{-/-}$ cells were cultured in 48-well plates and trans-fected with 200 ng of pcDNA3.1(+)-FLAG-CRBN-WT or 200 ng of pcDNA3.1 (+)-FLAG-CRBN-H353A and 15 ng of pcDNA3.1(+)-AGIA-SALL4. After incu-bation for 6 h, the cells were treated with DMSO (0.1%), thalidomide, or 5HT in culture medium at the indicated concentration for 18 h.

For experiments using SALL4 mutants, HEK293T-*CRBN*$^{-/-}$ cells were cultured in 48-well plates and transfected with 200 ng of pcDNA3.1(+)-FLAG-CRBN-WT and 15 ng of pcDNA3.1(+)-AGIA-SALL4-WT or 15 ng of pcDNA3.1(+)-AGIA-SALL4-mutant. After incubation for 6 h, the cells were treated with DMSO (0.1%), thalidomide, or 5HT in culture medium at the indicated concentration for 18 h.

For experiments using mutant IKZF1, the HEK293T-*CRBN*$^{-/-}$ cells were cultured in 48-well plates and transfected with 200 ng of pcDNA3.1(+)-FLAG-CRBN-WT and 15 ng of pcDNA3.1(+)-AGIA-IKZF1-WT or 15 ng of cDNA3.1 (+)-AGIA-IKZF1-mutant. After incubation for 6 h, the cells were treated with DMSO (0.1%), thalidomide, or 5HT in culture medium at the indicated concentration for 18 h.

For experiments using 5HP, the HEK293T-*CRBN*$^{-/-}$ cells were cultured in 48-well plates and transfected with 200 ng of pcDNA3.1(+)-FLAG-CRBN-WT, 15 ng of pcDNA3.1(+)-AGIA-SALL4-WT, and 15 ng of pcDNA3.1(+)-Myc-IKZF1-WT. After incubation for 6 h, the cells were treated with DMSO (0.1%), thalidomide, or 5HT in culture medium at the indicated concentration for 18 h.

For all experiments, the cells were lysed by boiling in 1× sample buffer (62.5 mM Tris-HCl, pH 6.8, 2% sodium dodecyl sulfate (SDS), and 10% glycerol) containing 5% 2-mercaptoethanol.

**Immunoblot analysis**. Protein lysates were separated by SDS-polyacrylamide gel electrophoresis and transferred onto polyvinylidene difluoride membranes (Milli-pore). After the membranes were blocked using 5% skim milk (Megmilk Snow Brand) in TBST (20 mM Tris-HCl, pH 7.5, 150 mM NaCl, and 0.05% Tween-20) at room temperature for 1 h, the appropriate antibody was used. Immobilon (Milli-pore) or ImmunoStar LD (FUJIFILM Wako Pure Chemical Corporation) was used as the HRP substrate, and the luminescence signals were detected using an Ima-geQuant LAS 4000 mini (GE Healthcare). To perform reprobing, stripping solution (FUJIFILM Wako Pure Chemical Corporation) was used, and the membranes were reblocked using 5% skim milk in TBST. The immunoblot data were analyzed by using ImageJ (version 2.0.0-rc-43/1.50e).

**Reporting summary**. Further information on research design is available in the Nature Research Reporting Summary linked to this article.

## Data availability
Coordinates and structure factors are available in the PDB under accession numbers 7BQV for the SALL4–CRBN complex with (*S*)-5HT and 7BQU for the SALL4–CRBN complex with (*S*)-thalidomide. The unprocessed scans of immunoblots are provided in a Source data file provided with this paper. Otherwise, the datasets generated and/or analyzed during the current study are available from the corresponding author on reasonable request. PDB coordinates used in this study are as follows: 4TZ4, 5YJ0, 5YJ1, 6H0F, and 6UML. Source data are provided with this paper.

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

## Acknowledgements

This research was supported by the Platform Project for Supporting Drug Discovery and Life Science Research (Basis for Supporting Innovative Drug Discovery and Life Science Research (BINDS)) from AMED under Grant Number JP20am0101077 (T.S., T.M.) and JP19am0101071 (support number 2290), a Grant-in-Aid for Scientific Research on Innovative Areas (JP16H06579 for T.S.) from the Japan Society for the Promotion of Science (JSPS), JSPS KAKENHI (JP17J08477 for S.Y., JP19H03218 for T.S., and JP17H06112 for N.S.), and Takeda Science Foundation. We appreciate Prof. Toshiya Senda, Dr. Yusuke Yamada, and the beamline staff for supporting the synchrotron radiation experiments performed with the AR-NE3A beamline in the Photon Factory (Tsukuba, Japan) (2019RP-36).

## Author contributions

M.T., T.S., and T.M. conceived and designed the project. H.F. performed biochemical studies and structural analyses with the support of T.M. and Y.M. S.Y. performed biochemical and cell-based experiments. T.H. and N.S. synthesized thalidomide and its derivatives. A.A. constructed the protein expression system for the structural analyses. H.F., S.Y., T.S., and T.M. wrote the original manuscript. M.T., T.S., and T.M. edited the manuscript.

## Competing interests

The authors declare no competing interests.
