## [Peer Review File · Nature Communications]

REVIEWER COMMENTS

Reviewer #1 (Remarks to the Author):

SALL4 and Ikaros are amongst the many “neo-morphic” substrates of the E3 ligase cereblon that are recruited by immunomodulatory drugs (IMiDs) such as thalidomide, pomalidomide and lenalidomide. While Ikaros was identified as a major on-targets of IMiDs therapeutic activity, particularly in multiple myeloma, SALL4 was recently implicated in the undesired teratogenic activity of thalidomide – although others have identified a different target (p63) for this, so this remains unsettled. Thalidomide and other IMiDs are metabolized in vivo by CYPs and chemically hydroxylated, with one such metabolite being the 5-OH phthalimide analogue (5HT)

Here the authors solved a new crystal structure of CRBN thalidomide binding domain (TBD) bound to (S)-thalidomide or its 5HT analogue, and the second zinc-finger (ZF2) of SALL4. This structure is related to one published recently by a team at Cellgene, which contained pomalidomide bound instead, and that the authors duly reference (ref. 38). An Alpha-Screen bead proximity assay is used to monitor recruitment of SALL4 to CRBN in the presence of compounds, with in one case ITC validation. Degradation assays with transfected plasmids are used to validate the impact of neo-morphic “glue-type” recruitment in the ternary complex to downstream proteasomal degradation. They found a water-mediated H-bond of the OH group with CRBN H353, validated biophysically and in cells. They then identify two important residues in SALL4 as forming tight interactions around the 5-OH group: Val441 and Arg418. By performing elegant residue-swap experiments, they establish that, unlike Thalidomide, 5HT preferentially recruits SALL4 ZF over Ikaros ZF. Finally, further mutations at the Val441 position are explored, which suggest that Ile and Thr this position are tolerated, and mutants bound as well as Val, while all other mutations significantly weakened binding. It would be interesting in future to test the extent to which ZFs containing Ile or Thr at this position e.g. WIZ ZF7 and OSR2 ZF1 might be also tightly and selectively glued into ternary complexes with CRBN by 5-OH IMiD analogues, and so could be off-target effects of these IMiDs metabolites. Also to aid more quantitative data, further work is warranted to characterize the thermodynamics and kinetics of these ternary complexes and dissect them apart from each other – for example using more information-rich biophysical assays e.g. by ITC and SPR which will provide more accurate quantification than Alpha-Screen signal intensity.

Nevertheless, together the presented results are solid and the experiments well designed and executed. This solid data-package strongly supports the authors claim that the SALL4 off-target degradation and so teratogenic effect is mainly due to IMiDs 5-OH in vivo metabolites. This is an important and timely finding because it furthers understanding of IMiD and neo-substrate specificity. It is significant also because the new structures and biophysical insights into the specificity could enable more rational design of more selective IMiDs and heterobifunctional degraders ie. PROTACs with reduced toxicity, allowing drug hunters to “dial-out” the unwanted off-target neo-substrate recruitment and so degradation effects. For these reasons, the paper warrants expeditious publication in my opinion, and I would be supportive of its publication in the journal, on the proviso that the authors duly address a few minor points of attention and concerns at revision.

- 1) Because the Cellgene structure is now published, however, the authors should scholarly compare and contrast their structures with that one, to highlight similarities and differences between the two. Also because there are now so many crystal structures of CRBN and its IMiD-mediated ternary complexes with neo-substrates, it seems redundant to describe known structural features in so much detail. The authors might want to focus their description mainly on the novel aspect of the structures.
- 2) The authors seem to imply that hydroxylation drives IMiD-mediated recruitment of SALL4. However, they provide no strong evidence for this claim. ITC titrations of ligand into CRBN WT and H353A mutant (Fig. 2b) should ideally be conducted also with thalidomide to provide comparison with its hydroxy metabolite.
- 3) For biophysical assays, the authors use a truncated CRBN TBD expressed recombinantly in E.coli.

They might want to comment on the extent to which this is a good proxy for the full-length protein, particularly in light of the induced protein-protein interactions within the ternary complex. For example, are there any missed interactions here because of using a truncated domain?

4) The green trace appears to be missing in Figure 3c, right panel?

I agree to waive anonymity
Alessio Ciulli

Reviewer #2 (Remarks to the Author):

An interesting paper showing a metabolite of thalidomide is shown to have an association with cereblon and SALL4 but which seems to differ in molecular interaction between thalidomide and other IMiDs.

This also indicates the possibility that teratogenic (and other) actions of thalidomide may in part also be caused by its breakdown products.

Other comments:

Abstract and Introduction

Line 19 and Line 62: I don't believe the binding of Thalidomide to SALL4 has been definitively shown to cause teratogenic effects in embryos, as yet. It is assumed this is potentially the case by both studies from 2018 and one study in bioRxiv that I am aware of – these studies were cell based and didn't directly test cereblon/sall4 and thalidomide interaction in embryos. Though one showed SALL4 is reduced in embryos after thalidomide exposure (as are many other genes).

Introduction

Line 33 and 34: Thalidomide affected many organs and tissues, upper and lower limbs, ears, eyes, face, cardiovascular system, kidneys, genitals, spinal column (and other tissues and organs) and the damage varied between survivors.

The introduction is primarily focused on thalidomide. As the focus of the paper is actually on one of the metabolites of thalidomide, 5HT, some more explanation as to why the authors wish to focus on 5HT might be helpful. This is further relevant as the authors cite on several occasions throughout the manuscript an article in BioRxiv (citation number 32) and use it to support several key statements yet appears to be very similar in findings to this manuscript.

Some information on how the authors study differs from that of citation number 32 would be helpful.

Results

Experiments are detailed and Figures are of good quality. There are a lot of Supplementary Figures and several are referred to multiple times in manuscript – I wonder if some of these could be added to the existing four Figures?

Discussion

Line 249: authors suggest CRBN and SALL4 causes teratogenicity. See earlier points. Can the authors detail in what species and what damage was seen? Is the damage caused by thalidomide and 5HT similar or different?

Line 262: more explanation on why the authors can state their data will help design new compounds with few adverse effects. All the data is biochemical and not tested in embryos.

Could the authors show if the interactions they are demonstrating biochemically occur in embryos treated with thalidomide and 5-HT and at relevant timepoints to be causing associated damage?

We have thoroughly revised our manuscript according to the comments of the reviewers. The following are point-by-point responses to the comments, and the revised portions of the manuscript are marked in red.

Reviewer #1 (Prof. Alessio Ciulli):

[Comment 1]

SALL4 and Ikaros are amongst the many “neo-morphic” substrates of the E3 ligase cereblon that are recruited by immunomodulatory drugs (IMiDs) such as thalidomide, pomalidomide and lenalidomide. While Ikaros was identified as a major on-targets of IMiDs therapeutic activity, particularly in multiple myeloma, SALL4 was recently implicated in the undesired teratogenic activity of thalidomide – although others have identified a different target (p63) for this, so this remains unsettled. Thalidomide and other IMiDs are metabolized in vivo by CYPs and chemically hydroxylated, with one such metabolite being the 5-OH phthalimide analogue (5HT)

Here the authors solved a new crystal structure of CRBN thalidomide binding domain (TBD) bound to (S)-thalidomide or its 5HT analogue, and the second zinc-finger (ZF2) of SALL4. This structure is related to one published recently by a team at Cellgene, which contained pomalidomide bound instead, and that the authors duly reference (ref. 38). An Alpha-Screen bead proximity assay is used to monitor recruitment of SALL4 to CRBN in the presence of compounds, with in one case ITC validation. Degradation assays with transfected plasmids are used to validate the impact of neo-morphic “glue-type” recruitment in the ternary complex to downstream proteasomal degradation. They found a water-mediated H-bond of the OH group with CRBN H353, validated biophysically and in cells. They then identify two important residues in SALL4 as forming tight interactions around the 5-OH group: Val441 and Arg418. By performing elegant residue-swap experiments, they establish that, unlike thalidomide, 5HT preferentially recruits SALL4 ZF over Ikaros ZF. Finally, further mutations at the Val441 position are explored, which suggest that Ile and Thr this position are tolerated, and mutants bound as well as Val, while all other mutations significantly weakened binding. It would be interesting in future to test the extent to which ZFs containing Ile or Thr at this position e.g. WIZ ZF7 and OSR2 ZF1 might be also tightly and selectively glued into ternary complexes with CRBN by 5-OH IMiD analogues, and so could be off-target effects of these IMiDs metabolites. Also to aid more quantitative data, further work is warranted to characterize the thermodynamics and kinetics of these ternary complexes and dissect them apart from each other – for example using more information-rich biophysical assays e.g. by ITC and SPR which will provide more accurate quantification than Alpha-Screen signal intensity.

Nevertheless, together the presented results are solid and the experiments well designed and executed. This solid data-package strongly supports the authors claim that the SALL4 off-target degradation and so theratogenic effect is mainly due to IMiDs 5-OH in vivo metabolites. This is an important and timely finding because it furthers understanding of IMiD and neo-substrate specificity. It is significant also because the new structures and biophysical insights into the specificity could enable more rational design of more selective IMiDs and heterobifunctional degraders ie. PROTACs with reduced toxicity, allowing drug hunters to “dial-out” the unwanted off-target neo-substrate recruitment and so degradation effects. For these reasons, the paper warrants expeditious publication in my opinion, and I would be supportive of its publication in the journal, on the proviso that the authors duly address a few minor points of attention and concerns at revision.

[Reply]

Thank you for the careful and thorough reading of our manuscript and for pointing out the major findings of our study. We have improved some expression in the revised manuscript according to your helpful comments and using your suitable phrases as follows.

Page 2, line 18 (Abstract)

p63 and PLZF were identified as other targets related to the undesired teratogenic activity of thalidomide. Therefore, we have added “in part” into the sentence “Thalidomide and its derivatives exert not only therapeutic effects as immunomodulatory drugs (IMiDs) but also adverse effects such as teratogenicity, which are due in part to different C2H2 zinc-finger (ZF) transcription factors, IKZF1 (or IKZF3) and SALL4, respectively.”

Page 2, line 27 (Abstract)

“the C2H2 ZF-type neosubstrate selectivity” has been changed to “the C2H2 ZF-type “neo-morphic” substrate (neosubstrate) selectivity”.

Page 2, lines 29 and 30 (Abstract)

“the molecular glue-like E3 ligase modulators” has been changed to “the “glue-type” E3 ligase modulators”.

Page 3, lines 44 and 45

“The neomorphic E3 ligase activity” has been changed to “The “neo-morphic” E3 ligase activity”.

Page 3, lines 48 and 49

“the CRL4^{CRBN}-dependent ubiquitination of neosubstrates” has been changed to “the CRL4^{CRBN}-dependent ubiquitination of “neo-morphic” substrates (neosubstrates)”.

Page 4, lines 63 and 64

We have added “implicated as one of the teratogenic candidates” to the sentence “Recently, spalt-like transcription factor 4 (SALL4), which is involved in foetal limb development and has a strong genetic link to embryopathies²⁶⁻²⁹, has been demonstrated as a C2H2 ZF-type neosubstrate for CRL4^{CRBN} and implicated as one of the teratogenic candidates of IMiDs³⁰⁻³².”

Page 4, line 76

We have added “by preventing unwanted off-target degradation” to the sentence “The mechanistic insights into the C2H2 ZF-type neosubstrate selectivity of thalidomide metabolites promote the understanding of the pharmaceutical actions of these IMiDs, which is required to develop thalidomide derivatives with reduced adverse effects by preventing unwanted off-target degradation.”

Page 13, lines 285 and 286

“the molecular glue-like E3 ligase modulators” has been changed to “the “glue-type” E3 ligase modulators”.

Page 15, lines 322–325

We have changed the sentence “Thus, our results provide an emerging conceptual framework for more effective design of IMiDs that selectively degrade C2H2 ZF-containing proteins through CRL4^{CRBN} with few adverse effects.” to “Thus, our results provide an emerging conceptual framework for more effective design of IMiDs with reduced off-target degradation and few adverse effects by selectively degrading C2H2 ZF-containing proteins through CRL4^{CRBN}.”

[Comment 2]

Because the Cellgene structure is now published, however, the authors should scholarly compare and contrast their structures with that one, to highlight similarities and differences between the two. Also because there are now so many crystal structures of CRBN and its IMiD-mediated ternary complexes with neo-substrates, it seems redundant to describe known structural features in so much detail. The authors might want to focus their description mainly on the novel aspect of the structures.

[Reply]

We have rearranged the section “Structure of the 5HT-mediated SALL4-CRBN complex” by adding the explanation of structural similarities and differences of (S)-5HT and (S)-thalidomide-mediated complex as compared with pomalidomide-mediated one as follows.

Page 6, lines 119–123

The binding mode of SALL4 ZF2 is similar to that in the reported crystal structure of pomalidomide-

mediated SALL4-CRBN-DDB1 complex³⁸, in which there is no contact of SALL4 ZF2 with regions other than the TBD (Fig. 2a). The position of β -hairpin loop relative to the CRBN TBD is completely aligned in the structures of three complexes mediated by 5HT, thalidomide or pomalidomide.

Page 7, lines 127–139

On the other hand, the Y355 side chain of the CRBN TBD, which is positioned near the β -hairpin loop of SALL4 ZF2, shows the different orientation from the pomalidomide-mediated complex (Fig. 2a). The residue is also located at the interface with the N-terminal domain (NTD) of CRBN, and the interaction Y355A mutation decreases the pomalidomide-mediated SALL4-CRBN interaction³⁸. Thus, the changes in the side-chain orientation may be due to the use of the truncated TBD and modulate the interaction between SALL4 ZF2 and the CRBN TBD. The other side-chain alteration was observed in the H378 residue of CRBN (Fig. 2a). In the complex structures, the H378 side chain is positioned in close proximity to the phthalimide moiety of (*S*)-5HT and (*S*)-thalidomide but far from that of pomalidomide. Since the 4-amino group of pomalidomide directs to the H378 residue, the differences in its side-chain orientation may be due to the structural modification of these compounds.

Page 7, lines 140–145

The overall orientation of SALL4 ZF2 is slightly tilted in the (*S*)-5-HT and (*S*)-thalidomide-mediated structures as compared with the pomalidomide-mediated structure (Fig. 2a), whereas this difference in orientation does not appear to affect the interaction between SALL4 ZF2 and the CRBN TBD. On the other hand, the improved resolution of the complex structure in this study shows novel evidence of some hydrogen-bonding networks on the interface between SALL4 ZF2 and the CRBN TBD.

To visually support these descriptions, we have added the new figures as Fig. 2a.

We would like to provide structural information needed for readers to understand this manuscript easily. We have simplified the description for the overall structure of CRBN TBD and SALL4 ZF2 within each sentence as follows.

Page 6, lines 105–111

The CRBN TBD adopts a twisted β -sheet (β 3- β 4- β 8- β 7- β 6- β 5) anchoring a β -hairpin (β 1 and β 2) with a zinc ion (Zn^{2+}) that binds with four cysteine residues (C323, C326, C391 and C394) in two CXXC motifs (Fig. 1c, Supplementary Fig. 2). On the other hand, SALL4 ZF2 shows a typical structure of C2H2 ZF domains consisting of a β -hairpin (β 1' and β 2') and an α -helix (α 1'), which are connected through Zn^{2+} binding with the conserved CXXC and HXXXH motifs (C412, C415, H428 and H432) (Fig. 1c, Supplementary Fig. 2).

[Comment 3]

The authors seem to imply that hydroxylation drives IMiD-mediated recruitment of SALL4. However, they provide no strong evidence for this claim. ITC titrations of ligand into CRBN WT and H353A mutant (Fig. 2b) should ideally be conducted also with thalidomide to provide comparison with its hydroxy metabolite.

[Reply]

We have obtained the ITC data showing the direct interaction of (*S*)-thalidomide with the wild-type and H353A mutant of CRBN TBD by changing the protein and ligand concentrations so as to improve exothermic signals (see the Materials and Methods section; page 19, lines 434–437). A buffer constituent “HEPES-NaOH” for ITC reaction has been corrected to “sodium phosphate” (page 19, line 431). The ITC data have been merged with those for (*S*)-5HT in Fig. 3d. (*S*)-5HT binds to the CRBN TBD with a K_D value of $0.76 \pm 0.20 \mu\text{M}$, and this affinity was decreased by an H353A mutation ($2.28 \pm 0.10 \mu\text{M}$). In contrast, the K_D values of (*S*)-thalidomide were almost the same toward the wild-type ($4.00 \pm 0.36 \mu\text{M}$) and H353A mutant ($4.43 \pm 0.16 \mu\text{M}$) of the CRBN TBD. These results indicate that (*S*)-5HT has a higher binding affinity toward CRBN TBD than (*S*)-thalidomide with the major contribution of the additional hydrogen-bond formation with the H353 side chain. We think that the enhanced binding of (*S*)-5HT toward CRBN TBD can partially explain the more profound 5HT-mediated formation of the SALL4-CRBN complex than that mediated by thalidomide. The description has been added on page 9, lines 183–191.

[Comment 4]

For biophysical assays, the authors use a truncated CRBN TBD expressed recombinantly in *E. coli*. They might want to comment on the extent to which this is a good proxy for the full-length protein, particularly in light of the induced protein-protein interactions within the ternary complex. For example, are there any missed interactions here because of using a truncated domain?

[Reply]

The structure of IMiD-mediated ternary complex using the full-length CRBN is very informative, but the resolution of the reported crystal structures is comparatively low, e.g. 3.58 \AA for CRBN-pomalidomide-SALL4 ZF2 complex and 3.25 \AA for CRBN-pomalidomide-IKZF1 ZF2 complex. Since the major objective in this study was the action of 5-hydroxy group of 5-HT in the complex formation induced between CRBN and SALL4, we utilized the truncated TBD instead of the full-length CRBN to obtain the high-resolution crystal structure that is required to determine the position of 5-hydroxy group in the ternary complex with SALL4 ZF2 clearly. As described in responses to the Specific Comment 1, SALL4 ZF2 does not contact any regions other than TBD. In addition, although only the Y355 residue of TBD may be related to the missed interaction because of using the truncated TBD, the positions of

the β -hairpin of SALL4 ZF2 and thalidomide compounds relative to the CRBN TBD are completely aligned between the complex structures using the truncated TBD and the full-length CRBN. Hence, the major interaction between SALL4 ZF2 and CRBN can be observed by using the truncated TBD as a good proxy for the full-length CRBN. On the other hand, some neosubstrates for PROTACs, such as BRD1 and GSPT1, interact not only with TBD but also with the NTD of CRBN. These structural evidences suggest that the use of the truncated TBD is restricted by an interaction mode of neosubstrate with CRBN. At least the interaction of a single C2H2 ZF domain including SALL4 ZF2 and IKZF1 ZF2 can be analyzed by using the truncated TBD. We have revised a part of the Discussion section using the related sentences as follows.

Page 14, line 314–page 15, line 322

We have changed “the constructs of the CRBN TBD and C2H2 ZF domains used for crystallization are helpful for the structure-guided drug design by enabling the visualization of the atomic structures of candidate compounds bound to the CRBN-C2H2 ZF complex.” to “In this study, the major interaction between SALL4 ZF2 and CRBN can be observed by using the truncated TBD as a good proxy for the full-length CRBN, whereas some neosubstrates for PROTACs, such as BRD4⁴² and GSPT1²³, interact not only with the TBD but also with the NTD of CRBN. These structural evidences suggest that the use of the truncated TBD is restricted by an interaction mode of neosubstrate with CRBN. At least the interaction of a single C2H2 ZF domain including SALL4 ZF2 and IKZF1 ZF2 can be analyzed by using the truncated TBD, which are helpful for the structure-guided drug design by enabling the visualization of the atomic structures of candidate compounds bound to the CRBN-C2H2 ZF complex.”

[Comment 5]

The green trace appears to be missing in Figure 3c, right panel?

[Reply]

The green trace was absolutely overlapped with the blue trace. The green trace has been placed in front of the blue trace in Fig. 4c and e (Fig. 3c and e in the original manuscript).

Reviewer #2

[Comment 1]

An interesting paper showing a metabolite of thalidomide is shown to have an association with cereblon and SALL4 but which seems to differ in molecular interaction between thalidomide and other IMiDs.

This also indicates the possibility that teratogenic (and other) actions of thalidomide may in part also be caused by its breakdown products.

[Reply]

Thank you for the helpful comments to improve our manuscript. We have revised the manuscript according to your comments as follows.

[Comment 2]

Abstract and Introduction

Line 19 and Line 62: I don't believe the binding of Thalidomide to SALL4 has been definitively shown to cause teratogenic effects in embryos, as yet. It is assumed this is potentially the case by both studies from 2018 and one study in bioRxiv that I am aware of – these studies were cell based and didn't directly test cereblon/sall4 and thalidomide interaction in embryos. Though one showed SALL4 is reduced in embryos after thalidomide exposure (as are many other genes).

[Reply]

There are several evidences strongly supporting that SALL4 is one of the most potential candidates involved in thalidomide teratogenicity such as: 1) SALL4 degradation is the most strongly induced by thalidomide compared with other neosubstrates; 2) there are genetic evidences which SALL4 downregulation cause several thalidomide-like phenotypes; and 3) SALL4 degradation occurs in only several species including human and rabbit and those species are highly sensitive to thalidomide. However, we previously reported PLZF as novel neosubstrate involved in thalidomide teratogenicity. Therefore, we fully agree to the reviewer's comment that CRBN-dependent neosubstrates involved in thalidomide teratogenicity are not only SALL4 but also other proteins. According to the reviewer's comment, we have mentioned the possibility that other proteins may be involved in teratogenicity in the revised manuscript as follows.

Page 2, line 18 (Abstract)

We have added “in part” into the sentence “Thalidomide and its derivatives exert not only therapeutic effects as immunomodulatory drugs (IMiDs) but also adverse effects such as teratogenicity, which are due in part to different C2H2 zinc-finger (ZF) transcription factors, IKZF1 (or IKZF3) and SALL4, respectively.”

Page 4, lines 63 and 64

We have added “implicated as one of the teratogenic candidates” into the sentence “Recently, spalt-like transcription factor 4 (SALL4), which is involved in foetal limb development and has a strong genetic link to embryopathies²⁶⁻²⁹, has been demonstrated as a C2H2 ZF-type neosubstrate for CRL4^{CRBN} and

implicated as one of the teratogenic candidates of IMiDs³⁰⁻³².”

Page 13, line 287–page 14, line 300

Thalidomide and its derivatives are typical drugs which show species specificity, and it has been reported that thalidomide does not show its teratogenic phenotypes in mice³⁰. Actually, the degradation of neosubstrates is mediated by CRBNs derived from highly sensitive species to thalidomide, including human and rabbit, but does not occur through mice CRBN^{19,30-32}. In addition, thalidomide low-sensitivity species, such as chicken, show the degradation of PLZF but not SALL4 by a thalidomide treatment³². Currently, it is thought that these differences of sensitivity to thalidomide result from differences of amino acid sequences of both CRBN and neosubstrates between the species^{19,30-32}. Therefore, these evidences and the molecular basis of 5HT-mediated formation of SALL4-CRBN complex strongly suggest that degradation of SALL4, PLZF and other proteins by both thalidomide and 5HT causes severe teratogenic phenotypes in highly sensitive species to thalidomide. However, many further researches using the highly sensitive species will be required to show whether what range and what tissues are affected by thalidomide and 5HT.

[Comment 3]

Introduction

Line 33 and 34: Thalidomide affected many organs and tissues, upper and lower limbs, ears, eyes, face, cardiovascular system, kidneys, genitals, spinal column (and other tissues and organs) and the damage varied between survivors.

The introduction is primarily focused on thalidomide. As the focus of the paper is actually on one of the metabolites of thalidomide, 5HT, some more explanation as to why the authors wish to focus on 5HT might be helpful. This is further relevant as the authors cite on several occasions throughout the manuscript an article in BioRxiv (citation number 32) and use it to support several key statements yet appears to be very similar in findings to this manuscript.

Some information on how the authors study differs from that of citation number 32 would be helpful.

[Reply]

We appreciate your kind suggestion about the effect range of thalidomide and that we should describe why we focused on 5HT. We understand that thalidomide causes severe teratogenicity in not only limbs but also many tissues and organs, such as heart, kidney, and ears. As according to the reviewer's comment, we have modified the explanation on teratogenicity range in introduction section (Page 3, lines 33 and 34). In our BioRxiv article (ref. 32), we revealed that 5HT has neosubstrate selectivity between SALL4 and IKZF1 and that 5HT has high potential for the degradation of SALL4. Because

of these differences in feature of 5HT compared with thalidomide, we focused on the structural bases for the selective proteasomal degradation of SALL4 induced by 5HT, which has been revealed in this study. According to the reviewer's comment, we have added the explanation about why we focused on 5HT as a new paragraph in the Introduction section as follows.

Page 4, lines 65–76

Thalidomide is mainly modified with 5-hydroxylation of the phthalimide moiety or 5'-hydroxylation of the glutarimide moiety through the action of cytochrome P450 isozymes^{33–35}. Recently, it has been reported that 5-hydroxythalidomide (5HT) has distinct neosubstrate selectivity between IKZF1 and SALL4³², and that 5HT induces degradation of SALL4 but not IKZF1³². Furthermore, 5HT induces more stronger degradation of SALL4 than thalidomide³². Therefore, it is predicted that 5HT contribute to thalidomide teratogenicity caused by SALL4 degradation. The molecular basis of its selectivity and degradation strength, however, has not been elucidated yet. The mechanistic insights into the C2H2 ZF-type neosubstrate selectivity of thalidomide metabolites will promote the understanding of the pharmaceutical actions of these IMiDs, which is required to develop thalidomide derivatives with reduced adverse effects by preventing unwanted off-target degradation.

[Comment 4]

Results

Experiments are detailed and Figures are of good quality. There are a lot of Supplementary Figures and several are referred to multiple times in manuscript – I wonder if some of these could be added to the existing four Figures?

[Reply]

Figure arrangement in the original manuscript was not comfortable for the readers. According to your comments, we have moved some Supplementary Figures as figures in main text as follows.

- 1) Supplementary Fig. 3 (original) has been merged to the figures newly prepared by the Reviewer 1's comments as Fig. 2 in the revised manuscript.
- 2) Supplementary Fig. 6 (original) has been added to Fig. 3 (Fig 2 in the original manuscript).
- 3) Supplementary Fig. 9 (original) has been added to Fig. 5 (Fig. 4 in the original manuscript), and the related sentence "The 5-hydroxylation of pomalidomide impairs the degradation of IKZF1 but not SALL4 through the interaction with CRBN as well as (*S*)-5HT (Fig. 5b–d)." has been added in the Results section (page 12, lines 256–258).

[Comment 5]

Discussion

Line 249: authors suggest CRBN and SALL4 causes teratogenicity. See earlier points. Can the authors detail in what species and what damage was seen? Is the damage caused by thalidomide and 5HT similar or different?

[Reply]

We thank you for pointing out important comments about species specificity and damage in embryos caused by thalidomide and 5HT.

In previous reports, it was shown that rabbit SALL4 is degraded by rabbit CRBN in the presence of thalidomide (ref. 30). On the other hand, in the case of mouse, chicken, and zebrafish, thalidomide cannot induce both interaction and degradation of SALL4 through their intrinsic CRBN because of difference in the amino acid sequence of CRBN between those species and human or rabbit (refs. 30–32). Based on these evidences, we believe that SALL4 degradation happens in highly sensitive species to thalidomide, such as human, primate, and rabbit, and that teratogenicity in low-sensitivity species, such as chicken, is caused by degradation of other proteins including PLZF. Probably, in highly sensitive species to thalidomide, degradation of SALL4, PLZF, and other proteins causes strong teratogenic phenotypes.

We previously reported that 5HT and thalidomide showed similar teratogenic phenotypes in chick limb bud, whereas the degradation of chicken SALL4 cannot be induced by both thalidomide and 5HT (ref. 32). In addition, it is still unknown whether similar phenotypes are observed in highly sensitive species to thalidomide. Given that 5HT more strongly induced interaction and degradation of SALL4 as shown in this study, it is predicted that 5HT treatment causes more strong teratogenicity through SALL4 degradation in those species. However, because further many *in vivo* studies are required to reveal those mysteries, we believe that it is beyond on scope of this study and we will address in the future research.

According to the reviewer's comment, we have added the explanation about species specificity and what damage by thalidomide in the Discussion section as follows.

Page 13, line 275–283

In recent studies, several proteins, such as SALL4 and PLZF, are reported to be the neosubstrates involved in thalidomide teratogenicity^{30–32}. Therefore, it is thought that several neosubstrates play functional roles in thalidomide teratogenicity, and SALL4 is one of the most potential candidates involved in thalidomide teratogenicity due to both genetic evidence of SALL4 and sensitivity of SALL4 degradation to thalidomide. Furthermore, 5HT induced degradation of both SALL4 and PLZF but not IKZF1 (Fig. 4c-f)³², and 5HT induces SALL4 degradation more strongly than thalidomide

(Fig. 3f). Recently, we showed that 5HT induced similar teratogenic phenotypes to thalidomide in chicken limb bud³².

Page 13, line 287–page 14, line 300

Thalidomide and its derivatives are typical drugs which show species specificity, and it has been reported that thalidomide does not show its teratogenic phenotypes in mice³⁰. Actually, the degradation of neosubstrates is mediated by CRBNs derived from highly sensitive species to thalidomide, including human and rabbit, but does not occur through mice CRBN^{19,30–32}. In addition, thalidomide low-sensitivity species, such as chicken, show the degradation of PLZF but not SALL4 by a thalidomide treatment³². Currently, it is thought that these differences of sensitivity to thalidomide result from differences of amino acid sequences of both CRBN and neosubstrates between the species^{19,30–32}. Therefore, these evidences and the molecular basis of 5HT-mediated neosubstrate degradation of CRBN strongly suggest that the degradation of SALL4, PLZF and other proteins by both thalidomide and 5HT causes severe teratogenic phenotypes in highly sensitive species to thalidomide. However, many further researches using the highly sensitive species will be required to show whether what range and what tissues are affected by thalidomide and 5HT.

[Comment 6]

Line 262: more explanation on why the authors can state their data will help design new compounds with few adverse effects. All the data is biochemical and not tested in embryos.

Could the authors show if the interactions they are demonstrating biochemically occur in embryos treated with thalidomide and 5-HT and at relevant timepoints to be causing associated damage?

[Reply]

We thank you for pointing out an important suggestion about in vivo experiments using embryo. As replied in Comment 5, in chicken embryos, 5HT and thalidomide show similar teratogenic phenotypes in limb bud (ref. 32) but SALL4 is not degraded in chicken embryos. Therefore, many in vivo experiments using highly sensitive species to thalidomide, rabbit or monkey, will be required to reveal whether strong degradation of SALL4 by 5HT causes more strong phenotypes. But this is beyond the scope of our present study and we will address this problem in the future. However, the evidence that rabbit SALL4 was degraded in rabbit embryo in the presence of thalidomide (ref. 30) and the structural bases for the selective degradation of SALL4 by 5HT shown in this study strongly suggest that 5HT more strongly induces the protein degradation in embryos than thalidomide. Furthermore, 5HP, which is produced by 5-hydroxylation modification of pomalidomide that is administrated currently for multiple myeloma therapy, also induces degradation of SALL4 but not IKZF1. Therefore, we propose that avoidance of the impact of 5-hydroxylation modification on the phthalimide moiety of IMiDs may reduce thalidomide teratogenicity affected by SALL4 degradation. According to the reviewer's

comment, we have added the discussion on teratogenic phenotypes by thalidomide and 5HT in the revised manuscript (page 13, line 287–page 14, line 300) and have changed the sentence “Therefore, further reduction in SALL4-mediated adverse effects requires the design of thalidomide derivatives based on the C2H2 ZF-type neosubstrate selectivity that depends on the 5-hydroxylation modification of the phthalimide moiety.” to “Therefore, avoidance of the impact of 5-hydroxylation modification on the phthalimide moiety of IMiDs may reduce thalidomide teratogenicity affected by SALL4 degradation. One of the approaches is the design of thalidomide derivatives based on the C2H2 ZF-type neosubstrate selectivity that depends on the 5-hydroxylation modification of the phthalimide moiety.” (page 14, lines 307–311).

Your kind and prompt judgement about this paper would be greatly appreciated.

REVIEWERS' COMMENTS:

Reviewer #1 (Remarks to the Author):

The authors have done a very good job at addressing my concerns and all reviewers comments. The revised manuscript now provides a more balanced discussion with due comparisons with the related Celgene structure. The new ITC data included strengthen the claim that 5HT (but not thalidomide itself) preferentially degrades SALL4.

Overall, I congratulate the authors on this study that will be an important addition to the field of protein degradation and thalidomide mode of action.

Reviewer #2 (Remarks to the Author):

I thank the authors for the detailed revisions to the manuscript. The majority of the comments i raised have been addressed.

I have only a few minor comments on the revised manuscript:

Title needs modification to include the precise metabolite name

Line 61 – is embryonic a better term than foetal - as limbs develop and fully form in the embryo stage of development (up to 8 weeks) and then 'just' lengthen in the foetal stage

Line 139 – I am unsure what the phrase 'slightly tilted' refers too

Line 274 – recently p63 was identified as having a role in thalidomide teratogenicity. Could the authors add this in the introduction and discussion?

Line 287 – the species differences in thalidomide action in mice was reported and known earlier than the reference cited.

Line 297 – agreed but I think it important to also confirm that the molecular interactions proposed are demonstrated to cause damage in-vivo and confirm if it's the cause of all the damage or parts of the damage as well as how the damage actually comes about.

We have thoroughly revised our manuscript according to the comments of Reviewer #2. The following are point-by-point responses to the comments, and the revised portions of the manuscript are shown using the 'track changes' feature in Word.

Reviewer #2:

[Comment 1]

I thank the authors for the detailed revisions to the manuscript.
The majority of the comments i raised have been addressed.

I have only a few minor comments on the revised manuscript:

[Reply]

We appreciate your further helpful comments. We have revised the manuscript according to your comments as follows.

[Comment 2]

Title needs modification to include the precise metabolite name

[Reply]

We have revised the title to “Structural bases of IMiD selectivity that emerges by 5-hydroxythalidomide”.

[Comment 3]

Line 61 – is embryonic a better term than foetal - as limbs develop and fully form in the embryo stage of development (up to 8 weeks) and then 'just' lengthen in the foetal stage

[Reply]

We have corrected the term “foetal” to “embryonic” on page 4, line 71.

[Comment 4]

Line 139 – I am unsure what the phrase ‘slightly tilted’ refers too

[Reply]

We have changed “The overall orientation of SALL4 ZF2 is slightly tilted” to “The overall orientation of SALL4 ZF2 to the CRBN TBD is slightly different”.

[Comment 5]

Line 274 – recently p63 was identified as having a role in thalidomide teratogenicity. Could the authors add this in the introduction and discussion?

[Reply]

We have stated that p63 is implicated as a teratogenic factor of IMiDs and a neosubstrate for CRBN in the Introduction and Discussion sections as follows.

Page 4, lines 63–64

Recently, spalt-like transcription factor 4 (SALL4), PLZF and p63 have been implicated as teratogenic candidates of IMiDs^{26–29}.

Page 13, lines 282–284

In recent studies, SALL4, PLZF and p63 have been reported to be the neosubstrates involved in thalidomide teratogenicity^{26–29}. Therefore, it is thought that several neosubstrates play functional roles in thalidomide teratogenicity.

As related to these statements, we have cited the following reference as ref. 29.

29. Asatsuma-Okumura, T. *et al.* p63 is a cereblon substrate involved in thalidomide teratogenicity. *Nat. Chem. Biol.* **15**, 1077–1084 (2019).

[Comment 6]

Line 287 – the species differences in thalidomide action in mice was reported and known earlier than the reference cited.

[Reply]

We have corrected the related citation to “Fratta, I. D., Sigg, E. B. & Maiorana, K. Teratogenic effects of thalidomide in rabbits, rats, hamsters, and mice. *Toxicol. Appl. Pharmacol.* **7**, 268–286 (1965).” as ref. 40.

[Comment 7]

Line 297 – agreed but I think it important to also confirm that the molecular interactions proposed are demonstrated to cause damage in-vivo and confirm if it's the cause of all the damage or parts of the damage as well as how the damage actually comes about.

[Reply]

We agree with your suggestion and have modified the related sentence “However, many further researches using the highly sensitive species will be required to show whether what range and what tissues are affected by thalidomide and 5HT.” to “However, many further researches using the highly sensitive species will be required to show whether what range and what tissues are damaged by thalidomide and 5HT through the molecular interactions proposed in this study.” (page 14, lines 305–308).

Your kind and prompt judgement about this paper would be greatly appreciated.